# AOEB: Benchmarking Agent-Oriented Multimodal Embeddings

**Xin Zhang** [1 2 †]  **Jiaxin Xu** [1]  **Mengjia Zhou** [1]  **Xinping Zhao** [1]  **Yinghui Li** [3]  **Di Yin** [3]  **Xing Sun** [3]  **Meishan Zhang** [1]
**Baotian Hu** [1]  **Wenjie Li** [2]  **Min Zhang** [1]

## Abstract

LLM agents powered by retrieval and RAG are increasingly prevalent across research and applications. Embedding models play a critical role in these systems, particularly in embedding-based retrieval. However, current benchmarks for embeddings remain focused on general-purpose scenarios, which may fail to align well with the diverse and evolving needs of agentic applications. To close this gap, we introduce Agent-Oriented Embedding Benchmark (AOEB), a comprehensive evaluation suite dedicated to agent-centric retrieval for embedding models. AOEB is characterized by two key features: (1) Multi-Task, covering five essential capabilities for retrieval in LLM agents, including code, tool, reasoning, and memory retrieval; and (2) Multi-Modal, providing evaluation with both textual and visual data for each task category. We evaluate representative embedding models on AOEB and observe that they exhibit distinct strengths across different agent-oriented retrieval tasks. By curating AOEB, we aim to promote a move toward more practically oriented directions within the embedding community and foster further progress.

## 1. Introduction

The remarkable capabilities of large language models (LLMs) have driven the rapid development of LLM-powered autonomous agents (Shinn et al., 2023; Wang et al., 2024a), making them increasingly prevalent and transformative across both academic research and real-world applications (Park et al., 2023; OpenAI, 2025; Lu et al., 2025). To mitigate hallucinations and enhance domain-specific knowledge of agentic systems, information retrieval and retrieval-

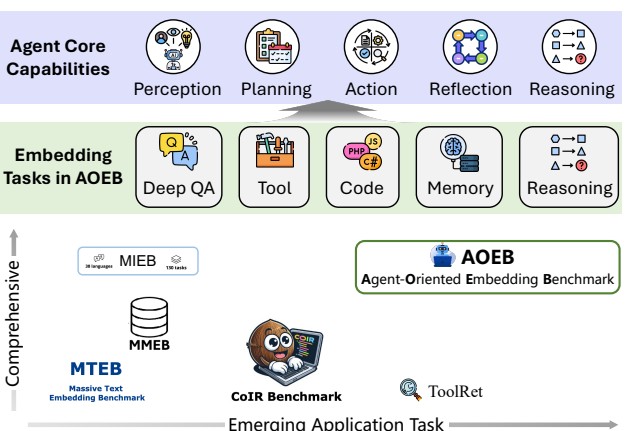

*Figure 1.* Design Illustration of AOEB, which is a comprehensive benchmark for agent-oriented retrieval. We derive five key meta-tasks that align with essential capabilities of LLM agents.

augmented generation (RAG) has emerged as critical techniques, enabling agents to ground their actions and responses in externally fetched information (Park et al., 2023; Wang et al., 2024a; Chen et al., 2025c). At the heart of these systems lie *embedding* models, which serve as the foundational cornerstone for in agent-centric retrieval tasks such as retrieving suitable tools from a vast collection (Shi et al., 2025), accessing pertinent memories from past interactions (Maharana et al., 2024), and searching code snippets for task execution (Wang et al., 2024a).

Current embedding evaluation benchmarks, *e.g.,* MTEB (Muennighoff et al., 2023) and BEIR (Thakur et al., 2021), have provided standardized frameworks for comparing embedding model performance. However, these benchmarks are predominantly designed for general-purpose scenarios, focusing on conventional tasks such as wikipedia retrieval, thereby showing a notable gap with the diverse, specialized, and dynamically evolving retrieval demands of agentic applications. They would fail to accurately measure how embedding models perform in agent-oriented retrieval tasks (Shi et al., 2025), making it difficult for users to choose the most effective model for their specific downstream needs. We believe that benchmarking of embedding models for the agent paradigm is crucial to guide the optimization of better models and to promote their adoption in agentic systems.

---

[†] Work done during an internship at Tencent. [1] Harbin Institute of Technology, Shenzhen [2] The Hong Kong Polytechnic University [3] Tencent Youtu Lab. Correspondence to: Meishan Zhang <zhangmeishan@hit.edu.cn>.

*Proceedings of the 43rd International Conference on Machine Learning*, Seoul, South Korea. PMLR 306, 2026. Copyright 2026 by the author(s).

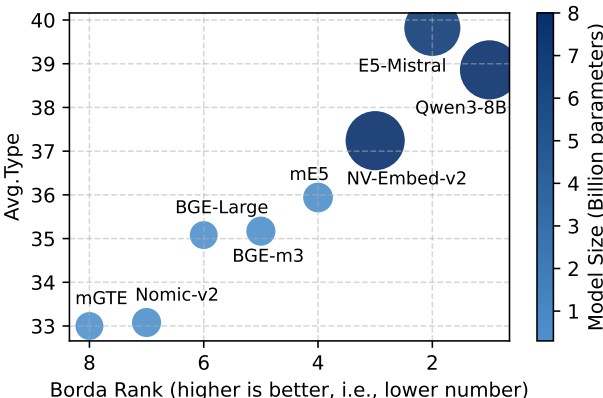

*Figure 2.* Model performance and Borda rank on AOEB-T. A more robust model has a higher rank than the average score, and is closer to the lower right corner of the coordinate system.

To bridge this gap, we introduce Agent-Oriented Embedding Benchmark (**AOEB**), which is tailored for assessing agent-oriented retrieval tasks (Figure 1). AOEB is featured by two key design principles: (1) Multi-Task Coverage: AOEB encompasses five capability dimensions crucial for agentic systems (§2.1), including deep QA (Chen et al., 2025c), code (Li et al., 2025), tool (Shi et al., 2025), reasoning (SU et al., 2025), and memory (Maharana et al., 2024) retrieval, which are employed in the key cycles of agent action and reflection. (2) Multi-Modal Evaluation: recognizing that modern agents must perceive and reason about the world through multiple modalities, AOEB provides both textual and visual data for evaluation within each task category (Van Landeghem et al., 2023; Liu et al., 2025b; Yang et al., 2025). By compiling existing datasets and creating new ones, AOEB thus offers a holistic and realistic evaluation of embedding models in scenarios that closely mirror real-world agentic applications.

We evaluate a diverse set of representative embedding models on AOEB (§4.1), including state-of-the-art multimodal models (Zhou et al., 2024; Chen et al., 2025a) and top-performing text-only models (Chen et al., 2024; Wang et al., 2024b; Lee et al., 2025). Our results reveal: First, we observe performance disparities across different agent-oriented tasks, with each model exhibiting distinct strengths and weaknesses[1]. Second, while larger models generally achieve stronger overall performance, we find that some less extensively trained models demonstrate superior generalization capabilities to novel tasks. We use the Borda rank (§2.3) to better capture model robustness across tasks, where certain models achieve high average scores but suffer from performance variance (Figure 2). Beyond aggregate metrics, we conduct further analysis (§4.3) to understand the benchmark and evaluated models. We hope the embedding

---

[1]For instance, code retrieval remains under-optimized for most embedding models, except for Jina-V4.

community could use AOEB to gain a holistic view of the current landscape, shifting the focus from overfitting general benchmarks toward new directions that are more potential.

**Notes after the acceptance:** This trend is already emerging, as evidenced by recent works that optimize embeddings for GUI agents (Zhang et al., 2025c) and search agents (Chen et al., 2026; Liu et al., 2026; Zhou et al., 2026), most of which are concurrent while complementary to our work.

In summary, we make three primary contributions:

- We introduce AOEB, the first comprehensive benchmark that evaluates embedding models across multiple skill dimensions and modalities specifically designed for retrieval in agent systems.
- We conduct an extensive evaluation of a range of representative models using AOEB, revealing their specific limitations and potential on these novel tasks.
- Through analysis and discussion, we provide concrete insights and directions for the future development of better agent-oriented embedding models.

Our benchmark and evaluation code is publicly available at `https://github.com/vec-ai/aoeb`.

## 2. AOEB: Agent-Oriented Embedding Bench

We present AOEB, a comprehensive benchmark designed to evaluate embedding models in diverse agent-oriented retrieval tasks with both textual and visual modalities. The overall design of AOEB is illustrated in Figure 1 & 3 and data composition is summarized in Table 1.

### 2.1. Task Design

Following recent trends like RTEB (Liu et al., 2025a), AOEB comprise only matching or retrieval tasks, which account for the vast majority of embedding applications. Concretely, for each dataset, we have a set of queries $Q = \{q_i\}$, a document collection $D = \{d_j\}$, and a set of relevance judgments $R = \{(q_i, d_j)\}$ that indicate which documents are relevant to which queries. The embedding model is used to encode both queries and documents into vector representations, and retrieval is performed by ranking documents based on their vector similarity to the query embeddings, obtaining a top-$K$ ranked list of retrieved documents for each query. The performance is then evaluated against $R$ using standard metrics such as Recall@$K$ and nDCG@$K$.

To compile AOEB, we establish three principles for selecting and constructing datasets: (1) Each dataset should be sufficiently meaningful and aligned with the practical application of its category. (2) Each dataset should present an appropriate level of difficulty; if most models have already achieved very high performance on a dataset, it is no longer suitable for evaluation. (3) The dataset size should be ap-

**AOEB**

**A**gent-**O**riented **E**mbedding **B**enchmark

⬳ Multi-Task          🖼️📄 Multi-Modal

**Deep QA** ⬳19
| Multi-Hop QA | Visual Doc |
| Deep Research | Technical Doc |

**Tool** ⬳4
| Web API | Tool Function |
| Customized App | Multimodal Tool |

**Reasoning** ⬳21
| Math Theorem | Psychology |
| Economics | Temporal |

**Code** ⬳10
| Code Search | Code QA |
| Chart to Code | Design to Code |

**Memory** ⬳10
| LoCoMo | LongMemEval |
| Multimodal Dialog | Skill Retrieval |

*Figure 3.* Overview of AOEB, which covers five key agent-oriented retrieval tasks across both textual and visual modalities, providing a comprehensive evaluation suite for embedding models in agentic applications. The detailed datasets under each type are listed in Table 1.

propriate: too few samples lead to non-robust evaluations, whereas excessively large datasets waste computation and annotation resources. The suitable scale is task-dependent[2]. We summarize five agent-oriented retrieval tasks as follows:

**Deep QA.** This task focuses on answering questions based on deep contextual understanding of documents. It represents the most fundamental form of retrieval, *i.e.,* finding the most relevant document for a question (query), yet recent advances in RAG and agent systems have introduced new characteristics, such as visual document retrieval (Ma et al., 2024; Faysse et al., 2025). Although many current search agents (Chen et al., 2025b) and deep research systems (OpenAI, 2025) do not rely on embedding models, opting instead for simple web search APIs, we contend that *embeddings remain essential*. On one hand, they serve as a core component of web search systems; on the other, they play a critical role in searching over local data (Zhao et al., 2024; Tan et al., 2026). The Deep QA meta-task in AOEB encompasses realistic (Thakur et al., 2025) and multi-hop (Trivedi et al., 2022) tasks, deep research (Chen et al., 2025c), as well as image-centric document retrieval (Liu et al., 2025b).

**Code Retrieval.** This meta-task evaluates embedding models on code-related search tasks, such as matching code snippets or functions that can be leveraged by agents to execute programming tasks. While code search has been widely studied in recent years, many existing benchmarks rely on data sourced from programming competitions or code documentations (Xie et al., 2023), which often lack relevance to real-world agentic coding applications. To enable more practical and production-oriented evaluation, we introduce code-to-code translation (Li et al., 2025) and newly curated tasks for web design- (Si et al., 2025) and chart-to-code (Yang et al., 2025) matching.

**Tool Retrieval.** This meta-task takes embedding models to retrieve appropriate tools or APIs based on user or model-generated queries, which enables agents to perform actions effectively. Compared to deep QA search and code retrieval, tool usage represents a fundamental skill for LLM-based agents, where tool retrieval acts as a critical first-stage filter in the tool selection and calling pipeline (Qin et al., 2024). AOEB includes tool retrieval tasks spanning API/function sets and applications (Shi et al., 2025), along with the demand of multimodal understanding (Fu et al., 2025).

**Reasoning-intensive Retrieval.** This meta-task tests the models to retrieve information that requires complex reasoning and logical deduction, which is crucial for agents to make informed decisions. To construct this evaluation, we select subsets from several recently established reasoning-intensive benchmarks, including knowledge and mathematical reasoning from BRIGHT (SU et al., 2025), temporal reasoning from RAR-B (Xiao et al., 2024), and multimodal reasoning from MRMR (Zhang et al., 2026).

**Memory Retrieval.** This task evaluates how well embedding models can retrieve relevant past interactions or memories, enabling agents to maintain context and continuity in their operations. It plays a vital role in context engineering (Mei et al., 2025), very long-context modeling (Maharana et al., 2024), and agent personalization (Tan et al., 2025). In AOEB, we implement this meta-task by introducing a new retrieval evaluation protocol variant, *i.e.,* `SubsetRetrieval`, which assigns each query $q_i$ to a specific subset of documents $D_{\text{subset}}$ within the whole collection, thereby constraining the retrieval to the memory of one single user. We integrate existing long-context memory retrieval datasets (Maharana et al., 2024; Wu et al., 2025) and contribute a novel multimodal one based on human-agent conversations (Xue et al., 2025). Detailed curation recipes are provided in the Appendix. In addition, we also inlude a new skill retrieval dataset SRA-Bench[3] (Su et al., 2026), which is also a critical task for agents to acquire useful procedural memories and capabilities.

---

[2]For example, QA datasets are naturally larger, while memory- and tool-oriented datasets tend to be smaller.

[3]SRA-Bench uses the standard retrieval evaluation protocol instead of `SubsetRetrieval`.

*Table 1.* AOEB evaluation datasets statistics. `T` and `I` denote text and image modalities, respectively. `IT` is the multimodal inputs with both image and text. `T/IT` means the document side can be either text or multimodal, *i.e.,* the collection are mixture of multiple modalities. *indicates the multimodal content (`IT`) of single input is image-text interleaved.

| Meta-Task | Dataset | Modality | #Subset | #Query | #Document |
|---|---|---|---|---|---|
| Deep QA | MuSiQue (Trivedi et al., 2022) | T→T | 1 | 2,417 | 21,100 |
| | BrowseComp-Plus (Chen et al., 2025c) | T→T | 1 | 830 | 100,195 |
| | FreshStack (Thakur et al., 2025) | T→T | 5 | 672 | 271,842 |
| | DUDE (Van Landeghem et al., 2023) | T→I | 1 | 496 | 27,955 |
| | MVRB (Liu et al., 2025b) | T/IT→I | 11 | 3,008 | 39,884 |
| Tool Retrieval | ToolRet (Shi et al., 2025) | T→T | 3 | 7,961 | 44,453 |
| | ReFocus (Fu et al., 2025) | IT→T | 1 | 3,600 | 10,479 |
| Code Retrieval | COIR (Li et al., 2025) | T→T | 8 | 58,368 | 389,120 |
| | Design2Code (Si et al., 2025) | T→I | 1 | 484 | 484 |
| | ChartMimic (Yang et al., 2025) | T→I | 1 | 2,400 | 2,400 |
| Reasoning-intensive | BRIGHT (SU et al., 2025) | T→T | 10 | 1,130 | 918,110 |
| | RAR-B (Xiao et al., 2024) | T→T | 7 | 13,824 | 44,032 |
| | MRMR (Zhang et al., 2026) | IT/I→T/IT* | 4 | 947 | 16,553 |
| Memory Retrieval | LoCoMo (Maharana et al., 2024) | T→T | 5 | 1,982 | 5,882 |
| | LongMemEval (Wu et al., 2025) | T→T | 1 | 500 | 51,572 |
| | SRA-Bench (Su et al., 2026) | T→T | 2 | 10,800 | 52,524 |
| | MMRC-R (Xue et al., 2025) | T→T/IT* | 2 | 5,360 | 5,730 |

## 2.2. Modality Composition

AOEB features both textual and visual data to reflect the multimodal nature of modern agentic systems, where agents must perceive and reason about the environment through multiple modalities (Wang et al., 2024a). Specifically, we categorize AOEB datasets based on the modality of the query and document into the following types: T→T (text-to-text), T→I (text-to-image), T→IT (text to image-text), IT→I (image-text to image), IT→T, and IT→IT. There are also some datasets, specifically retrieval collections, are mixed-modality, *e.g.,* MRMR documents are comprised of images and image-text multimodal content. Hence, we use notation like T/IT denote the modality mixture in collection. It is worth noting that some image-text content may provide multiple images per instance or even *image-text interleaved* sequences (depending on the original dataset, *e.g.,* MRMR (Zhang et al., 2026), MMRC-R (Xue et al., 2025)), which poses additional challenges for embedders to effectively represent the semantics.

**AOEB & AOEB-T.** Recognizing that many agentic applications and embedding models remain text-only, we also provide AOEB-T, the *text-only* subset of AOEB.

## 2.3. Metric & Ranking

Most datasets in AOEB employ standard retrieval metric nDCG@10 as the main metric by default. Current embedding benchmarks (Muennighoff et al., 2023; Jiang et al., 2025; Xiao et al., 2025b) often report average scores across all tasks to provide a quick overview of model performance and relative ranking. We presnet the overall average (both

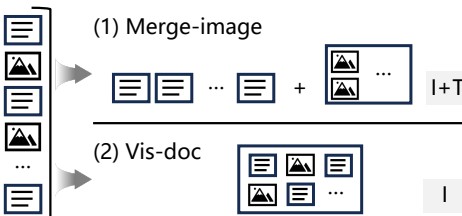

*Figure 4.* Illustration of the two approaches for adapting single-image models to interleaved inputs. Merge-image combines all images into one, while Visual-doc renders the entire interleaved document as a single image.

micro and macro) scores as well for reference. However, this approach might be in a dilemma of overlooking the robustness and consistency of models across different datasets. For instance, a model might achieve a high average score by excelling in some tasks while performing poorly in a few tasks, which could be detrimental in real-world applications if the model fails in critical scenarios. To provide a better overall ranking, we adopt the Borda count method (Enevoldsen et al., 2025), which aggregates rankings from individual datasets into a final model ranking. This approach takes into account the relative performance of models across tasks.

## 3. Evaluation Method

### 3.1. Embedding Models

We evaluate a diverse set of embedding models in AOEB, including both multimodal and text-only ones. We list the exact huggingface model identifiers in the Appendix §B.1.

**Multimodal Embedding.** We select representative mod-

*Table 2.* AOEB multimodal evaluation results. `Context` indicates the maximum input context length in our evaluation, not the real maximum context length of the model, which is much longer. We emphasize the best and second-best results in **bold** and underline.

| Model | Context | Size | Rank$_{Borda}$ | Avg.$_{Type}$ | Avg.$_{64}$ | Deep QA$_{19}$ | Tool$_4$ | Code$_{10}$ | Reasoning$_{21}$ | Memory$_{10}$ |
|---|---|---|---|---|---|---|---|---|---|---|
| VISTA | 512 | 0.2B | 7 | 30.58 | 25.27 | 15.03 | 35.37 | 38.21 | 15.05 | 49.24 |
| VLM2Vec | 8192 | 4B | 8 | 26.12 | 24.97 | 30.59 | 17.98 | 26.19 | 11.34 | 44.51 |
| GME$_{2B}$ | 8192 | 2B | 4 | 42.27 | 38.26 | 41.70 | 42.56 | 54.59 | 19.59 | 52.90 |
| GME$_{7B}$ | 8192 | 8B | 2 | 48.65 | 45.18 | 56.36 | **45.09** | 64.35 | 20.21 | 57.23 |
| mmE5 | 8192 | 11B | 5 | 39.97 | 38.22 | 47.72 | 30.60 | 55.42 | 18.33 | 47.75 |
| Jina-v4 | 8192 | 4B | 3 | 45.06 | 42.29 | 52.78 | 36.07 | 70.33 | 17.71 | 48.42 |
| VLM2Vec-v2 | 8192 | 2B | 6 | 38.06 | 36.49 | 47.29 | 24.58 | 53.04 | 14.01 | 51.39 |
| Qwen3VL$_{2B}$ | 8192 | 2B | 1 | **51.03** | **48.06** | **59.16** | 42.63 | **70.46** | **22.55** | **60.32** |

els that have demonstrated strong performance in universal multimodal retrieval: (1) encoder-based model: VISTA (Zhou et al., 2024) is the only one that produce a unified embedding space for both text and images; (2) multimodal LLM (MLLM)-based models: they are fine-tuned from MLLM backbones and naturally generated aligned multimodal embeddings. We include VLM2Vec (Jiang et al., 2025), VLM2Vec-V2 (Meng et al., 2026), GME (Zhang et al., 2025e), mmE5 (Chen et al., 2025a), Jina-v4 (Günther et al., 2025), and Qwen3-VL Embedding (Li et al., 2026).

**Text Embedding.** There exists a wide range of options, including English-specific, multilingual, and models of varying size scales (from 0.1B parameter encoders to 8B parameter decoders). While these models are optimized for general retrieval tasks, AOEB serves as an out-of-distribution evaluation for them. To identify models with greater potential, we select ones that have shown strong generalization capabilities on RTEB-English (Liu et al., 2025a), including (1) encoder-based: BGE-m3 (Chen et al., 2024), mE5 (Wang et al., 2024c), mGTE (Zhang et al., 2024), Nomic-v2 (Nussbaum & Duderstadt, 2025); (2) LLM-based: E5-Mistral (Wang et al., 2024b), NV-Embed-v2 (Lee et al., 2025), and Qwen3 Embedding (Zhang et al., 2025g). In addition, we also include the above multimodal models for text-only evaluation on AOEB-T.

### 3.2. Adaption of Interleaved Inputs

The evaluated open-source multimodal embedding models are designed to take only one image at a time. However, multi-image and image-text interleaved contents are inevitable (Zhang et al., 2025b; 2026). AOEB also contains such data, especially in reasoning-intensive (Zhang et al., 2026) and memory retrieval (Xue et al., 2025) tasks. To adapt single-image models to interleaved inputs, we employ two straightforward approaches: (1) *Merge-image*: all images in the interleaved input are merged into a single composite image (by resizing and concatenation), and then we pair it with the entire text content to form a single image-text input for embedding computation; (2) *Visual-doc*: we render

the entire interleaved content into a multimodal document and feed the screenshot of it as a single image input to the model. Figure 4 illustrates the idea of these two methods. We acknowledge that these are preliminary solutions, and call for more native models.

## 4. Experiments

### 4.1. Setting

We implement AOEB based on the MTEB (Muennighoff et al., 2023; Enevoldsen et al., 2025) toolkit 2.0 version, which provides a unified and extensible framework for retrieval and embedding. Since MTEB do not support retrieval collections that are mixture of multiple modality settings, we implement a new framework to evaluated datasets in this case, *i.e.,* MRMR (Zhang et al., 2026) and MMRC-R (Xue et al., 2025). For all evaluated models, we use their official Sentence Transformers (Reimers & Gurevych, 2019) implementations if available, or otherwise build upon their official code example to ensure correct usage. The max context length is set to 8192 for computational efficiency. We run all experiments in half-precision (float16 or bfloat16, depending on model specifications) on machines with 8 NVIDIA GPUs. For multimodal models in interleaved datasets, we apply the two adaptation approaches described in §3.2 and report the better results in Table 2 and Table 3. Other details such as processing and inputs are provided in the Appendix.

### 4.2. Overall Results

We present the overall evaluation results of representative models on AOEB in Table 2 and Table 3 (text-only).

**Multimodal.** We evaluate several top MLLM-based and encoder-based embedding models on the AOEB full set (Table 2). The extensively optimized `Qwen3VL-2B` (Li et al., 2026) achieves the best overall performance, demonstrating its strong generalization ability across diverse agent-oriented retrieval tasks. The encoder-based model `VISTA` (Zhou et al., 2024) lags behind the MLLM-based models, which

*Table 3.* AOEB-T text-only evaluation results of representative multimodal and text-only embedding models. `Context` indicates the maximum input context length in our evaluation, not the real maximum context length of the model, which is much longer. We emphasize the best and second-best results in **bold** and underline, respectively.

| Model | Context | Size | Rank$_{Borda}$ | Avg.$_{Type}$ | Avg.$_{43}$ | Deep QA$_7$ | Tool$_3$ | Code$_8$ | Reasoning$_{17}$ | Memory$_8$ |
|---|---|---|---|---|---|---|---|---|---|---|
| *Text Embedding Models* | | | | | | | | | | |
| BGE$_{Large}$ | 512 | 0.3B | 9 | 35.08 | 30.17 | 17.22 | 47.06 | 44.94 | 16.29 | 49.90 |
| BGE-m3 | 8192 | 0.5B | 7 | 35.17 | 31.14 | 16.80 | 44.27 | 47.43 | 18.95 | 48.37 |
| mE5 | 512 | 0.6B | 6 | 35.94 | 32.16 | 16.33 | 42.04 | 52.24 | **19.08** | 50.03 |
| mGTE | 8192 | 0.3B | 13 | 33.00 | 28.16 | 17.80 | 42.63 | 52.01 | 13.56 | 38.99 |
| Nomic-v2 | 8192 | 0.5B | 12 | 33.08 | 28.56 | 15.79 | 33.81 | 46.72 | 17.12 | 45.25 |
| E5$_{Mistral}$ | 8192 | 7B | 3 | 39.82 | 35.01 | 19.56 | 46.12 | 58.75 | 18.07 | 56.62 |
| NV-Embed-v2 | 8192 | 8B | 5 | 37.24 | 31.40 | 17.63 | 48.55 | 47.37 | 13.46 | **59.18** |
| Qwen3$_{8B}$ | 8192 | 8B | 2 | 38.86 | 34.45 | **27.29** | 37.27 | 57.00 | 15.62 | 57.11 |
| *Multimodal Embedding Models* | | | | | | | | | | |
| GME$_{2B}$ | 8192 | 2B | 8 | 36.90 | 31.68 | 13.97 | 49.81 | 59.33 | 16.61 | 44.77 |
| GME$_{7B}$ | 8192 | 8B | 4 | 39.67 | 34.23 | 13.16 | **52.47** | 64.75 | 17.84 | 50.12 |
| mmE5 | 8192 | 11B | 11 | 33.09 | 29.76 | 14.32 | 37.85 | 52.12 | 17.72 | 43.45 |
| Jina-v4 | 8192 | 4B | 10 | 37.58 | 32.28 | 17.32 | 47.24 | 67.92 | 15.39 | 40.01 |
| VLM2Vec-v2 | 8192 | 2B | 14 | 29.93 | 26.14 | 11.87 | 31.23 | 51.16 | 10.54 | 44.83 |
| Qwen3VL$_{2B}$ | 8192 | 2B | 1 | **42.70** | **37.39** | 22.57 | 49.52 | **68.21** | 18.91 | 54.29 |

might be attributed to its freezed text embedding space and limited model capacity. In addition, `mmE5` and `VLM2Vec` achieves unsatisfactory results, possibly due to their *General Embedding* training objective (Jiang et al., 2025), *i.e.,* optimized for classification, grounding *etc.* , which may not align well with the specific requirements of agent-oriented retrieval tasks. Overall, all models are still far from perfect, leaving significant room for improvement in agent-oriented multimodal retrieval tasks.

**Text-only.** In Table 3, we evaluate popular text embedding models (both encoder-based and LLM-based ones), and we also include multimodal models which claim strong text embedding capabilities (*i.e.,* strong performance on MTEB). `Qwen3VL-2B` is still the best model. Meanwhile, we observe that in the same parameter scale, there is no clear winner between the text-only models and multimodal ones, *e.g.,* `Qwen3`$_{8B}$ (Zhang et al., 2025g) slightly outperforms `GME`$_{7B}$ (Zhang et al., 2025e). This not only suggests that multimodal models can also provide strong text embeddings, but also indicates that *the agent-oriented retrieval tasks is like a out-of-distribution test for both types of models*, which are not specifically optimized for such scenarios. To this end, more dedicated training and optimization are needed to further enhance embedding quality for agent applications.

**Dilemma in Average Scores.** As stated in §2.3, AOEB reports dataset-level micro averages (Avg.$_{43}$), meta-task macro averages (Avg.$_{Type}$), and Borda ranks to provide a more comprehensive evaluation. We expect Borda ranking to better reflect overall robustness across diverse agent-

oriented retrieval tasks. In the text-only evaluation (Table 3), with 14 models in total, two cases illustrate why relying on a single average can be misleading. First, `GME`$_{7B}$ achieves a higher macro average than `Qwen3`$_{8B}$ (39.67 *v.s.* 38.86), while `Qwen3`$_{8B}$ attains a slightly higher micro average (34.45 *v.s.* 34.23). Borda resolves this tension and ranks `Qwen3`$_{8B}$ 2nd and `GME`$_{7B}$ 4th, favoring the model with stronger dataset-level performance. Second, `E5`$_{Mistral}$ and `GME`$_{2B}$ differ on both averages (Avg.$_{Type}$ 39.82 *v.s.* 36.90; Avg.$_{43}$ 35.01 *v.s.* 31.68), yet the gap is amplified in Borda ranking (3rd *v.s.* 8th). This suggests that `E5`$_{Mistral}$ is more consistent across datasets, whereas `GME`$_{2B}$ exhibits larger cross-task variance despite moderate macro scores. Together, these cases demonstrate that Borda ranking offers a more nuanced view of model performance on agent-oriented retrieval benchmarks than either average alone.

### 4.3. Analysis

**Performance Trends to General Benchmark (Figure 5).** We analyze the performance trends of the evaluated models in relation to general benchmarks, such as MTEB and other established retrieval tasks, to show how well these models generalize to agent-oriented scenarios. We collect the results of our evaluated models on MTEB (Muennighoff et al., 2023), RTEB-English (Liu et al., 2025a), and MMEB (Jiang et al., 2025) from their online leaderboards, as well as MIEB (Xiao et al., 2025b) from (Xiao et al., 2025a). Figure 5 illustrates the comparison results. We can see that the performance trends of models on AOEB differ from those on

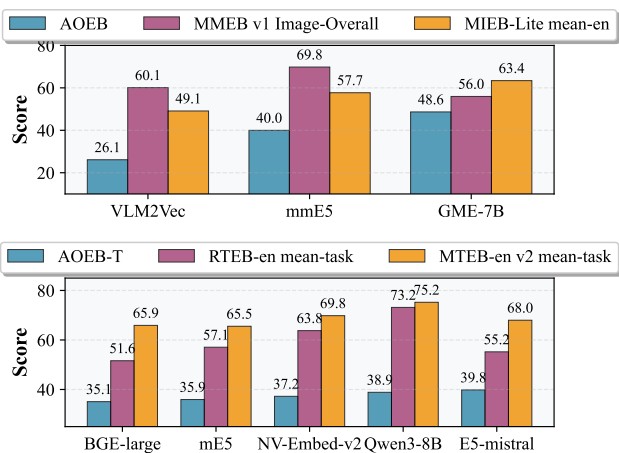

*Figure 5.* Model performance comparison between AOEB and general benchmarks. The upper sub-figure shows the results of multimodal embedding models (with MMEB and MIEB-lite), while the lower one presents text embedding models (MTEB-en-v2, RTEB-en). The models are ordered by performance on AOEB (from left to right, worse to better). We observe that models that perform well on general benchmarks do not necessarily excel in AOEB, indicating the unique challenges in AOEB.

general benchmarks. In AOEB multimodal, `VLM2Vec` < `mmE5` < `GME_7B`, while in MMEB `VLM2Vec` is not the worst. In text-only setting, model rankings of MTEB-en-v2 and RTEB-en are also different from AOEB-T. `E5-Mistral` show strong generalization in AOEB-T, while it is not the best in MTEB-en-v2 and RTEB-en. And some well-fitted models on general benchmarks even perform poorly on AOEB, such as `NV-Embed-v2`. These observations suggest that AOEB presents tests and characteristics that are not fully captured by existing benchmarks. And the specific optimization for agent-oriented retrieval tasks is necessary to improve generalization of new embedding models.

**Downstream Task Validation (Table 4).** To validate the practical implications of our benchmark findings, we include end-to-end experiments on downstream agent tasks that rely on retrieval, *e.g.,* FreshStack (Thakur et al., 2025) realistic document RAG and BrowseCompPlus (Chen et al., 2025c) deep research agent. As shown in Table 4, we select three representative embedding models, *i.e.,* one early LLM embedder, one latest LLM embedder and one multimodal embedder, from our evaluation and assess the corresponding downstream performance. We observe a strong correlation between the embedding performance on AOEB and the end-to-end RAG results, indicating that *better embedding models lead to improved retrieval quality and overall performance* in downstream applications. This also confirms that AOEB effectively reflects the capabilities of embedding models in practical agent-oriented retrieval applications. We believe that further advancements in embedding models, guided by AOEB, will directly translate to enhanced performance in

*Table 4.* Downstream task evaluation results on FreshStack RAG (Thakur et al., 2025) and BrowseComp+ (Chen et al., 2025c) deep research agent. `F.S.-R` is the retrieval performance on the FreshStack from our evaluation, while `F.S.` is the end-to-end RAG coverage (Thakur et al., 2025). `BC+` is the deep research accuracy via LLM judgment (Chen et al., 2025c)

| Model | AOEB | AOEB-T | F.S.-R | F.S | BC+-R | BC+ |
|---|---|---|---|---|---|---|
| Qwen3$_{8B}$ | - | 38.86 | 27.41 | 0.689 | 20.09 | 8.55 |
| Jina-v4 | 45.06 | 37.58 | 17.24 | 0.584 | 4.92 | 3.13 |
| E5$_{Mistral}$ | - | 39.82 | 19.46 | 0.553 | 8.72 | 4.78 |

real-world LLM agent systems.

**Interleaved Adaption Effectiveness (Table 5).** In the main results of AOEB, we adapt single-image models to interleaved inputs using two approaches: `Merge-image` and `Visual-doc`, and the better results are reported in Table 2. To further understand the effectiveness of these approaches, we present a detailed comparison in Table 5, showing the performance of both methods on interleaved datasets. We observe that models specifically optimized for visual document retrieval, such as `Jina-v4` (Günther et al., 2025) and `GME` (Zhang et al., 2025e), benefit significantly from the `Visual-doc` adaption approach, achieving substantial performance improvements over Merge-image. In contrast, models that are not trained on visual document retrieval data (*e.g.,* VISTA and VLM2Vec) tend to perform better with the `Merge-image` approach. Condisering the `Visual-doc` models, no matter optimized for this setting or not, generally outperform `Merge-image` ones in adaptation, we believe the *visual document representation is a more effective way to handle interleaved content* for current embedding models. The score difference (*i.e.,* `diff` in Table 5, `Visual-doc` - `Merge-image`) between the two methods highlights the varying adaptability of different models to interleaved inputs, where the positive and larger valuables are expected for better adaptation. As information-rich multimodal documents become more prevalent in agent applications, it is crucial to develop embedding models that can natively handle interleaved content effectively.

**Benchmark Runtime (Table 6).** We also report the runtime of our benchmark evaluation, which is an important practical consideration for researchers and practitioners. As shown in Table 6, the GPU hours of evaluating a single model on AOEB can range from a few hours to one hundred hours. We believe that the runtime is acceptable and align with our design principles in §2.1 since we could parallelize the evaluation across multiple devices.

## 5. Discussion

We believe the shift toward agentic scenarios is a natural step for embedding research.

*Table 5.* Comparison of interleaved input adaptation methods on AOEB interleaved datasets. `MI` and `VD` denote *Merge-image* and *Visual-doc* approaches, respectively. `Diff` indicates the performance difference (`VD - MI`). For visual document retrieval optimized models, *e.g.,* `Jina-v4` and `GME`, `VD` significantly outperforms `MI`. For models that are not trained on visual document retrieval data, the behavior is reversed.

| Dataset → Subset → Model ↓ | MRMR traffic MI/VD | MRMR theorem MI/VD | MMRC-R session MI/VD | Avg Diff |
|---|---|---|---|---|
| VISTA | 21.53/6.69 | 19.93/0.98 | 74.10/31.01 | -25.6 |
| VLM2Vec | 16.93/20.06 | 18.23/13.77 | 46.39/68.91 | 7.1 |
| GME$_{2B}$ | 33.12/37.54 | 25.43/26.45 | 82.14/83.50 | 2.3 |
| GME$_{7B}$ | 36.47/36.54 | 29.83/23.06 | 80.76/83.51 | -1.3 |
| mmE5 | 19.00/32.17 | 5.81/16.64 | 64.12/73.37 | 11.1 |
| Jina-v4 | 15.33/25.14 | 19.84/21.38 | 76.74/79.26 | 4.6 |
| VLM2Vec-V2 | 23.87/34.98 | 21.54/17.77 | 63.47/80.03 | 8.0 |
| Qwen3VL$_{2B}$ | 44.57/44.30 | 33.95/30.10 | 83.86/85.00 | -1.0 |

**Generalization.** We observe that both encoders and 7/8B decoders exhibit cases where newer models which perform better on general benchmarks are nonetheless weaker than older models they appear to surpass. This suggests that such models may be overfitting the evaluation distributions of standard benchmarks rather than learning more broadly applicable semantic representations. Future research should prioritize improving generalization to ensure robust performance across diverse and previously unseen tasks.

**Code Enhancement.** Code retrieval is critical in agent applications. In our evaluation, `Jina-v4` remains among the strongest on code retrieval, likely due to extensive code data in training. `Qwen3VL-2B` is comparable on these aggregates, which is also attributed to its large-scale optimization. Newer models may benefit from adopting similar training data strategies to strengthen their code retrieval capabilities.

**Modality.** Currently, AOEB includes only text and image modalities, which is a limitation. We acknowledge that audio and video are important modalities for agent applications. As an initial step, we focus on text and image because they are the most common and foundational. Future work should extend the benchmark to additional modalities to enable a more comprehensive assessment of embedding models in multimodal agent scenarios.

**Learning from Agent.** To build agent native retrieval embedding models, it is crucial to find effective supervision signals and data sources that are closely aligned with the needs of agents. Multi-turn agent trajectories are an increasingly practical supervision source for this purpose. Liu et al. (2026) combine local sub-query relevance with global answer correctness in contrastive training and co-evolve retriever and agent; Chen et al. (2026) embed reasoning traces with each query and use synthesized sub-query data; Zhou et al. (2026) derive labels from browse versus skip decisions

*Table 6.* Runtime statistics of representative models on AOEB. GPU hours on single NVIDIA A6000 are reported.

| Model | Size | Context | Type | Runtime (hours) |
|---|---|---|---|---|
| BGE-large | 0.3B | 512 | Text-only | 2.61 |
| BGE-m3 | 0.5B | 8192 | Text-only | 6.52 |
| Qwen3$_{8B}$ | 8B | 8192 | Text-only | 82.33 |
| VISTA | 0.2B | 512 | Multimodal | 16.77 |
| GME$_{2B}$ | 2B | 8192 | Multimodal | 29.48 |
| GME$_{7B}$ | 7B | 8192 | Multimodal | 80.16 |
| mmE5 | 11B | 8192 | Multimodal | 170.45 |
| Jina-v4 | 4B | 8192 | Multimodal | 97.05 |
| Qwen3VL$_{2B}$ | 2B | 8192 | Multimodal | 121.75 |
| VLM2Vec-v2 | 2B | 8192 | Multimodal | 43.77 |

and post-browse reasoning with relevance-aware weighting. Together, we could see that trajectory-mined utilities and signals should complement curated question–document corpora for the training of agent-oriented embeddings. Notably, these works typically evaluate embedding improvements through end-to-end search agent performance, as well as reductions in the number of search calls required by the agent. Such metrics are the most faithful evaluation criteria for agent-oriented retrieval systems. However, compared with conventional retrieval benchmarking, these indirect end-to-end evaluations are substantially more expensive and less scalable. We believe an interesting open question is whether proxy evaluation protocols could be developed to efficiently approximate downstream agent performance.

## 6. Related Work

**Benchmarking Embedding and Retrieval.** Evaluation of embedding and retrieval models remains a long-standing research topic (Enevoldsen et al., 2025; Zhang et al., 2025a). The BEIR benchmark (Thakur et al., 2021) focuses specifically on information retrieval across heterogeneous domains, providing zero-shot evaluation scenarios. Towards comprehensive evaluation, Massive Text Embedding Benchmark (MTEB) (Muennighoff et al., 2023) represents a landmark effort, encompassing 56 diverse tasks across retrieval, classification, clustering, and semantic textual similarity. MTEB has become the de facto standard for evaluating text embedding models. Recent works have extended MTEB to multilingual or multimodal settings, such as MMTEB (Enevoldsen et al., 2025), which incorporates hundreds of languages, as well as MMEB (Jiang et al., 2025) and MIEB (Xiao et al., 2025b), which include multimodal embedding tasks. However, the recent efforts in benchmarking embedding models gradually shift back to only retrieval tasks, such as RTEB (Liu et al., 2025a), recognizing that retrieval remains the most prevalent application of embeddings (Günther et al., 2025; Zhang et al., 2025d). Despite these advances, existing benchmarks primarily target general-purpose scenarios and do not adequately capture the unique demands in new

applications like LLM agents. Our work addresses this gap by introducing AOEB for agent-oriented retrieval tasks.

**Retrieval in LLM Agents.** The integration of retrieval mechanisms into LLM agents has emerged as a critical research direction for addressing knowledge limitations and hallucination issues. Agent-oriented retrieval systems face unique challenges compared to traditional search scenarios. Multi-hop reasoning tasks like deep research require agents to iteratively retrieve and synthesize information across multiple steps (Chen et al., 2025c), while multimodal agents must handle diverse input modalities including text, images, and structured data (Wang et al., 2024a). Recent work on tool-using agents (Wang et al., 2025) has demonstrated the importance of retrieval for grounding agent actions in external knowledge bases, where tool retrieval acts a pivotal role (Shi et al., 2025; Zhang et al., 2025f). However, most current agent systems rely on black-box APIs rather than optimized embedding models, limiting their adaptability and transparency (Chen et al., 2025c). There is a growing recognition of the need for embedding models that are specifically optimized for the unique demands of agent-oriented retrieval tasks. Zhang et al. (2025c) present the first attempt to optimize embedding models for agents, more specifically, GUI agents, where useful trajectories are retrieved to augment the action planning. Very recently, Liu et al. (2026) explore the joint optimization of embedding models and agents, demonstrating that better embeddings can lead to improved agent performance. Chen et al. (2026) and Zhou et al. (2026) propose methods for training embedding models which understand the reasoning traces of deep research agents, and also learn from agent interactions. Our work contributes to this area by providing systematic evaluation of embedding models specifically for agent-oriented scenarios, thereby guiding future research towards more effective embeddings for LLM agents.

## 7. Conclusion

We present AOEB, a comprehensive benchmark for evaluating embedding models in agent-oriented retrieval tasks across multiple skill dimensions, including deep QA, tool, code, reasoning and memory. Through the extensive evaluation of representative models, we reveal their specific limitations and potential in these novel tasks. We also conduct downstream agent evaluations to confirm that better AOEB scores lead to improved agent performance. Our analysis provides concrete insights and directions to guide the future development of more powerful agent-oriented embedding models. We hope that AOEB could serve as a usefule resource for the research community, fostering advances in embedding techniques tailored to the evolving needs of LLM agents and related applications.

## Acknowledgements

We thank the anonymous reviewers and chairs for their valuable feedback and suggestions. This work receives support from the Natural Science Foundation of China (under Grants 624B2048, 62336008), Research Grant Council of Hong Kong (PolyU/15205325), and the Shenzhen Basic Research Program (Grant No. JCYJ20240813105111016).

## Limitations

In summary, the limitations are threefold:

- Limited model coverage. We currently evaluate a set of representative and well-documented models, but some of the latest models are not yet included. We will continuously expand the benchmark to include newer models in MMEB and MIEB.

- Room for task expansion. The current benchmark can be further extended in several directions. First, multilinguality: most current tasks are in English, while modern embedding models are often multilingual. Second, task design: as agent-related scenarios evolve rapidly, new tasks such as agent trajectory retrieval and agent skill retrieval can be incorporated.

- Handling overfitting. At present, AOEB mainly mitigates overfitting by introducing diverse and newly constructed tasks. However, once AOEB is publicly released, the risk of overfitting remains. This can be further addressed by continuously updating the data and maintaining a portion of private test sets.

## Impact Statement

This paper aims to advance embedding and retrieval research toward scenarios that are better aligned with agentic applications. By introducing a benchmark for evaluating embedding models in agent-oriented retrieval tasks, we intend to provide the community with a comprehensive view of current model performance in these emerging settings. This can help reveal existing limitations and motivate future work toward more capable embedding techniques tailored for LLM agents and related applications.

At present, evaluations of embedding models in the community are largely centered on general-purpose retrieval. The intense competition for SOTA results has led many recent models to overfit existing benchmarks, reducing their ability to generalize to new scenarios. We encourage the community to refocus on generalization, enabling embedding models to better adapt to diverse task requirements and expanding the practical impact of embedding studies.

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

# Appendix

# A. AOEB Details

## A.1. Deep QA

**MuSiQue.** (Trivedi et al., 2022) is a challenging multihop QA dataset constructed by composing connected single-hop questions, resulting in 25K 2–4 hop questions that require genuine multi-step reasoning.

**BrowseComp-Plus.** (Chen et al., 2025c) is a controlled deep-research benchmark with a fixed curated corpus, enabling fair and reproducible evaluation of search-augmented LLMs.

**FreshStack.** (Thakur et al., 2025) is an automatically constructed IR benchmark framework that builds challenging, up-to-date datasets from technical corpora and community Q&A, offering realistic and difficult evaluation settings for retrieval and RAG systems.

**DUDE.** (Van Landeghem et al., 2023) is a practical Document AI benchmark built from diverse, real-world, visually rich documents, offering multi-task and multi-domain evaluation settings that better reflect real-world document understanding challenges.

**MVRB.** (Liu et al., 2025b) is a comprehensive benchmark for visualized information retrieval, evaluating how well models retrieve information from unified visual screenshots across diverse tasks and modalities.

## A.2. Tool Retrieval

**ToolRet.** (Shi et al., 2025) is a large-scale benchmark for tool retrieval, designed to evaluate how well models can find the right tools for agentic LLMs—a capability where existing IR models perform surprisingly poorly.

**ReFocus.** (Fu et al., 2025) can be seen as a benchmark where multimodal LLMs retrieve visual editing tools in sequence to perform structured image reasoning, testing their ability to select the right tools to arrive at the correct answer.

## A.3. Code Retrieval

**COIR.** (Li et al., 2025) is a comprehensive benchmark for code retrieval, covering ten datasets across seven domains and eight retrieval tasks, designed to evaluate how well models can find relevant code snippets or APIs from diverse sources.

**Design2Code.** (Si et al., 2025) tests how well models can retrieve the appropriate code or layout patterns to generate

*Table 7.* Summary of datasets in AOEB.

| Meta-Task | Dataset | Subset | Modality | #Query | #Document | Main-Metric |
|---|---|---|---|---|---|---|
| Deep QA | MuSiQue | ans-dev | T→T | 2,417 | 21,100 | NDCG@10 |
| Deep QA | BrowseCompPlus | gold | T→T | 830 | 100,195 | NDCG@10 |
| Deep QA | FreshStack | angular | T→T | 129 | 117,288 | NDCG@10 |
| Deep QA | FreshStack | godot | T→T | 99 | 25,482 | NDCG@10 |
| Deep QA | FreshStack | langchain | T→T | 203 | 49,514 | NDCG@10 |
| Deep QA | FreshStack | laravel | T→T | 184 | 52,351 | NDCG@10 |
| Deep QA | FreshStack | yolo | T→T | 57 | 27,207 | NDCG@10 |
| Deep QA | DUDE | - | T→I | 496 | 27,955 | NDCG@10 |
| Deep QA | MVRB | ComposedScreenshotRetrievalKnowledgeRelation | IT→I | 100 | 1,011 | NDCG@10 |
| Deep QA | MVRB | ComposedScreenshotRetrievalNewsToWiki | IT→I | 101 | 1,010 | NDCG@10 |
| Deep QA | MVRB | ComposedScreenshotRetrievalProductDiscovery | IT→I | 107 | 1,012 | NDCG@10 |
| Deep QA | MVRB | ComposedScreenshotRetrievalWikiToProduct | IT→I | 100 | 969 | NDCG@10 |
| Deep QA | MVRB | ScreenshotRetrievalChartRetrieval | IT→I | 200 | 5,000 | NDCG@10 |
| Deep QA | MVRB | ScreenshotRetrievalDocRetrieval | IT→I | 200 | 5,000 | NDCG@10 |
| Deep QA | MVRB | ScreenshotRetrievalNewsRetrieval | IT→I | 491 | 5,401 | NDCG@10 |
| Deep QA | MVRB | ScreenshotRetrievalPaperRetrieval | IT→I | 503 | 5,021 | NDCG@10 |
| Deep QA | MVRB | ScreenshotRetrievalProductRetrieval | IT→I | 496 | 5,436 | NDCG@10 |
| Deep QA | MVRB | ScreenshotRetrievalRepoRetrieval | IT→I | 510 | 5,024 | NDCG@10 |
| Deep QA | MVRB | ScreenshotRetrievalSlideRetrieval | IT→I | 200 | 5,000 | NDCG@10 |
| Tool Retrieval | ToolRet | code | T→T | 1,749 | 3,794 | NDCG@10 |
| Tool Retrieval | ToolRet | customized | T→T | 982 | 3,367 | NDCG@10 |
| Tool Retrieval | ToolRet | web | T→T | 5,230 | 37,292 | NDCG@10 |
| Tool Retrieval | ReFocus | - | IT→T | 3,600 | 10,479 | NDCG@10 |
| Code Retrieval | COIR | AppsRetrieval | T→T | 3,800 | 9,000 | NDCG@10 |
| Code Retrieval | COIR | CodeFeedbackMT | T→T | 13,000 | 66,000 | NDCG@10 |
| Code Retrieval | COIR | CodeFeedbackST | T→T | 31,000 | 156,000 | NDCG@10 |
| Code Retrieval | COIR | CodeTransOceanContest | T→T | 446 | 1,000 | NDCG@10 |
| Code Retrieval | COIR | CodeTransOceanDL | T→T | 180 | 816 | NDCG@10 |
| Code Retrieval | COIR | CosQA | T→T | 500 | 21,000 | NDCG@10 |
| Code Retrieval | COIR | StackOverflowQA | T→T | 2,000 | 20,000 | NDCG@10 |
| Code Retrieval | COIR | SyntheticText2SQL | T→T | 6,000 | 105,851 | NDCG@10 |
| Code Retrieval | Design2Code | - | T→I | 484 | 484 | NDCG@10 |
| Code Retrieval | ChartMimic | - | T→I | 2,400 | 2,400 | NDCG@10 |
| Reasoning Intensive | BRIGHT | aops | T→T | 111 | 188,002 | NDCG@10 |
| Reasoning Intensive | BRIGHT | biology | T→T | 103 | 57,883 | NDCG@10 |
| Reasoning Intensive | BRIGHT | earth-science | T→T | 116 | 121,850 | NDCG@10 |
| Reasoning Intensive | BRIGHT | economics | T→T | 103 | 50,736 | NDCG@10 |
| Reasoning Intensive | BRIGHT | psychology | T→T | 101 | 53,347 | NDCG@10 |
| Reasoning Intensive | BRIGHT | robotics | T→T | 101 | 62,469 | NDCG@10 |
| Reasoning Intensive | BRIGHT | stackoverflow | T→T | 117 | 108,939 | NDCG@10 |
| Reasoning Intensive | BRIGHT | sustainable-living | T→T | 108 | 61,346 | NDCG@10 |
| Reasoning Intensive | BRIGHT | theoremqa-questions | T→T | 194 | 188,002 | NDCG@10 |
| Reasoning Intensive | BRIGHT | theoremqa-theorems | T→T | 76 | 23,839 | NDCG@10 |
| Reasoning Intensive | RAR-B | TempReasonL1 | T→T | 4,000 | 12,504 | NDCG@10 |
| Reasoning Intensive | RAR-B | TempReasonL2Context | T→T | 5,397 | 15,787 | NDCG@10 |
| Reasoning Intensive | RAR-B | TempReasonL2Fact | T→T | 5,397 | 15,787 | NDCG@10 |
| Reasoning Intensive | RAR-B | TempReasonL2Pure | T→T | 5,397 | 15,787 | NDCG@10 |
| Reasoning Intensive | RAR-B | TempReasonL3Context | T→T | 4,426 | 15,664 | NDCG@10 |
| Reasoning Intensive | RAR-B | TempReasonL3Fact | T→T | 4,426 | 15,664 | NDCG@10 |
| Reasoning Intensive | RAR-B | TempReasonL3Pure | T→T | 4,426 | 15,664 | NDCG@10 |
| Reasoning Intensive | MRMR | Theorem | IT→T/IT | 579 | 14,257 | NDCG@10 |
| Reasoning Intensive | MRMR | Design | IT→T/IT | 88 | 700 | NDCG@10 |
| Reasoning Intensive | MRMR | Traffic | IT→T/IT | 80 | 796 | NDCG@10 |
| Reasoning Intensive | MRMR | Negation | I→T | 200 | 800 | NDCG@10 |
| Memory Retrieval | LoCoMo | adversarial | T→T | 446 | 5,882 | NDCG@10 |
| Memory Retrieval | LoCoMo | multi-hop retrieval | T→T | 282 | 5,882 | NDCG@10 |
| Memory Retrieval | LoCoMo | open-domain knowledge | T→T | 92 | 5,882 | NDCG@10 |
| Memory Retrieval | LoCoMo | single-hop retrieval | T→T | 841 | 5,882 | NDCG@10 |
| Memory Retrieval | LoCoMo | temporal reasoning | T→T | 321 | 5,882 | NDCG@10 |
| Memory Retrieval | LongMemEval | m(user side utterance) | T→T | 500 | 51,572 | NDCG@10 |
| Memory Retrieval | SRA | brief | T→T | 5,400 | 26,262 | NDCG@10 |
| Memory Retrieval | SRA | full | T→T | 5,400 | 26,262 | NDCG@10 |
| Memory Retrieval | MMRC-R | round | T→I/T | 2,680 | 5,321 | NDCG@10 |
| Memory Retrieval | MMRC-R | session | T→IT | 2,680 | 409 | NDCG@10 |

webpages from visual inputs like screenshots.

**ChartMimic.** (Yang et al., 2025) evaluates how well models can retrieve the correct chart code given textual instructions.

### A.4. Reasoning-intensive Retrieval

**BRIGHT.** (SU et al., 2025) is a reasoning-intensive retrieval benchmark that evaluates how well models can retrieve relevant documents for complex queries requiring deep logical and domain-specific reasoning, beyond simple keyword or semantic matching.

**RAR-B.** (Xiao et al., 2024) is a reasoning-as-retrieval benchmark that evaluates how well retriever models can handle reasoning-intensive tasks, testing their ability to retrieve information that supports complex problem-solving rather than simple semantic matching.

**MRMR.** (Zhang et al., 2026) is a multimodal, reasoning-intensive retrieval benchmark, testing models on complex image-text reasoning. The traffic and theorem subsets contain documents with multiple images and image-text interleaved instances.

### A.5. Memory Retrieval

**LoCoMo.** (Maharana et al., 2024) is a dataset of very long-term dialogues, originally designed to study long-range consistency and temporal reasoning, which we adapt for memory retrieval to evaluate models' ability to retrieve relevant past interactions.

**LongMemEval.** (Wu et al., 2025) is a benchmark for evaluating long-term memory in chat assistants, adapted for memory retrieval to test models' ability to retrieve relevant information across multiple sessions, covering tasks like temporal reasoning, knowledge updates, and multi-session question answering.

**MMRC-R.** (Xue et al., 2025) is a multimodal conversation dataset adapted for memory retrieval, including text and images, across sustained multi-turn dialogues. We construc two retrieval settings: round-level and session-level. The documents in session-level are with multiple images and image-text interleaved.

**SRA-Bench.** (Su et al., 2026) is a benchmark for skill retrieval in LLM agents, which we adapt for memory retrieval as the skills are procedural memory that agents need to retrieve and execute. We setup two retrieval settings: brief and full, where the former only contains the skill description while the latter also includes the full skill markdown content as the document text.

## B. Experimental Details

### B.1. Evaluated Models

We evaluate a set of representative embedding models from both multimodal and text-only categories, covering a range of architectures and training paradigms. Here we list the exact model identifiers from huggingface for reproducibility.

- VISTA (BAAI/bge-visualized)
- VLM2Vec (TIGER-Lab/VLM2Vec-LoRA)
- GME-2B (Alibaba-NLP/gme-Qwen2-VL-2B-Instruct)
- GME-7B (Alibaba-NLP/gme-Qwen2-VL-7B-Instruct)
- mmE5 (intfloat/mmE5-mllama-11b-instruct)
- Jina-v4 (jinaai/jina-embeddings-v4)
- BGE-Large (BAAI/bge-large-en-v1.5)
- BGE-m3 (BAAI/bge-m3)
- mE5 (intfloat/multilingual-e5-large-instruct)
- mGTE (Alibaba-NLP/gte-multilingual-base)
- Nomic-v2 (nomic-ai/nomic-embed-text-v2-moe)
- E5-Mistral (intfloat/e5-mistral-7b-instruct)
- NV-Embed-v2 (nvidia/NV-Embed-v2)
- Qwen3-8B (Qwen/Qwen3-Embedding-8B)
- Qwen3VL-2B (Qwen/Qwen3-VL-Embedding-2B)
- VLM2Vec2 (VLM2Vec/VLM2Vec-V2.0)

### B.2. Processing of MMRC

For conversational datasets, we randomly sample documents to form a memory sub-corpus whose size is drawn from a discrete uniform distribution from minimum 7 to maximum 50 with mean 24. Retrieval is then performed within each sub-corpus. We build 17 sub-corpora for MMRC; the corresponding qrels are re-computed so that only documents inside the same sub-corpus can be judged relevant. In the qrels labels, the "1" indicates relevant while "0" indicates irrelevant but in the sub-corpus.

**Query Generation.** MMRC (Xue et al., 2025) is a multimodal conversation dataset without retrieval queries. To build a dataset for evaluating memory retrieval, we use a two-stage pipeline that automatically generates recall-style queries from image content, spatiotemporal cues, and conversational context.

**Stage I: Spatiotemporal metadata generation..** Given a raw image, a vision model extracts scene-level features and produces aligned temporal and spatial metadata, including the event time, scene location, and an environmental description. This stage maps visual observations to a unified time–space anchor that better matches how humans organize episodic memories, providing a consistent reference for subsequent memory construction.

**Stage II: Recall-oriented query generation..** An LLM

then takes the image, the generated spatiotemporal metadata, and the surrounding multi-turn dialogue as joint inputs to produce recall-type retrieval queries. The model conditions on visual scene content, dialogue semantics, and the time-space anchor so that each query reflects realistic memory-triggering patterns rather than isolated image captioning.

Query generation follows five principles:

1. Embed time and location as event anchors so recall is grounded in a specific scene, not detached visual content alone.
2. Set the question time later than the event time to simulate post-hoc recall rather than contemporaneous questioning.
3. Prefer relative temporal expressions (*e.g.,* "last week," "last month," "that afternoon") over explicit calendar dates, discouraging string matching and stressing temporal association.
4. Use multi-turn dialogue to supply situational background and user behavior, so queries leverage event context beyond what is visible in the image.
5. Emphasize environmental, situational, and experiential details rather than simple factual QA, aligning queries with how humans trigger episodic memories.

Together, these stages turn raw image-centric interactions into recall queries linked across time, space, and dialogue context, yielding a more realistic long-term memory retrieval setting and a stronger test of temporal reasoning and memory recall.

### B.3. All Results

We list the detailed result of each data subsets in Table 9 (Deep QA), Table 10 (Code & Tool), Table 11 (Reasoning) and Table 12 (Memory).

*Table 8.* Data examples from AOEB datasets.

| Task Name | Query Example | Document Example |
|---|---|---|
| Deep QA MuSiQue | Who is the spouse of the Green performer? | Miquette Giraudy (born 9 February 1953, Nice, France) is a keyboard player and vocalist, best known for her work in Gong and with her partner Steve Hillage. She and Hillage currently form the core of the ambient band System 7. In addition to her performances in music, she has also worked as an actress, film editor and writer. In each role, she has used different stage names. |
| Deep QA BrowseComp-Plus | Please tell me the name of the learning institution that fits the following criteria: A. In 2002, it held a three-day event from Thursday to Saturday. Its activities centered primarily around showing support to a group of people. B. In 2003, it held its graduation ceremony on the fourth Sunday of a particular month. C. In 2022, an article was published on this educational institution's website about a trip for certain year levels of students from a particular academic department to gather samples of plants. D. Seven days after the article discussed in Criterion C was published, an academic division of this learning establishment organized a ceremony to pay tribute to the management of a bank with the support of a top university official. E. The country's capital city, as of 2023, is where the learning institution is situated. | — title: Arwa University holds annual cultural activities [Archives:2002/20/Local News] date: 2002-05-13 — Arwa University holds annual cultural activities [Archives:2002/20/Local News] BY ABDUH AL-SABRI YEMEN TIMES STAFF Under the sponsorship of the president Ali Abdullah Saleh and under the motto (Jerusalem is Ours), the Queen Arwa University held its Fifth Cultural Week (FCW) on May 9-11 2002, which was coincided with the Palestine Intifadha against the Zionist atrocities in the occupied land. The 3-day FCWs activities focused mainly on the Palestinian support. Symposiums, political and cultural lectures were delivered to show solidarity with people in Palestine. Al-Afeef Cultural Foundation had a vital participation in making the FCW a success. Dr. Abdulaziz Abdulghani, the Shura Council Chairman highly appreciated the efforts exerted by the Queen Arwa University and making of the FCW a success in packing the Palestinians up. Our brothers in Palestine have been fighting tooth and nails for the sake of their legal right which represents a true exemplary of a resounding victory of strong Palestinian will, the Shura Council Chairman highly praised Palestinians. In the same context, the T.V. Sector Chairman, Abdulghani al-Shameri said that the Queen Arwa University has done its spare no efforts to hold such activities for the benefit of the Palestinians. We are hoping that other universities will follow the example of the Queen Arwa University, in highlighting the Palestinian issue to the public, al-Shameri said to the Yemen Times reporter. On the sideline of the FCWs activities, a bazaar was allotted to support the Palestinian Intifadha. Poetic sessions soiree was delivered by the Dr. Abdulaziz al-Makaleh and Sulaiman al-Aisi in conjunction with al-Afeef Cultural Foundation. In addition to this, several exhibitions shows were presented by the Arab and foreign embassies in our country. The FCW was attended by the Shura Council Chairman, Abdulaziz Abdu |

| Task Name | Query Example | Document Example |
|---|---|---|
| Deep QA FreshStack | Is there any sense already for using signals in Angular?According to RFC3 signal-based components with change detection strategy based fully on signals are planned as next thing to be released. So as of now, with zone-based change detection strategy, is there any sense of using signals over the traditional way of setting values to class' properties? Will signals' dependency tree eg. gain performance in zone-based components? | ## Releasing APIs before they're fully stable
The Angular team may occasionally seek to release a feature or API without immediately including this API in Angular's normal support and deprecation category. You can use one of two labels on such APIs: Developer Preview and Experimental. APIs tagged this way are not subject to Angular's breaking change and deprecation policy.
Use the sections below to decide whether a pre-stable tag makes sense.
### Developer Preview
Use "Developer Preview" when: * The team has relatively high confidence the API will ship as stable. * The team needs additional community feedback before fully committing to an exact API shape. * The API may undergo only minor, superficial changes. This can include changes like renaming or reordering parameters, but should not include significant conceptual or structural changes.
### Experimental
Use "Experimental" when: * The team has low-to-medium confidence that the API should exist at all. * The team needs additional community feedback before deciding to move forward with the API at all. * The API may undergo significant conceptual or structural changes. * The API relies on a not-yet-standardized platform feature. |
| Deep QA DUDE | Instruct: You need to retrieve evidence from a PDF page to address the question. Query: What is the minimum wage Agricultural Employees on January 1, 2020? | 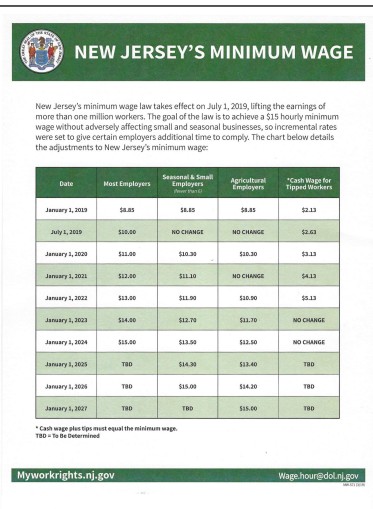 |

| Task Name | Query Example | Document Example |
|---|---|---|
| Tool Retrieval ToolRet | Instruction: Given a 'language identification' task, retrieve tools that process audio inputs to accurately determine the language being spoken, utilizing models designed for audio classification and language detection purposes.Query: You are building a virtual global tour guide that can identify languages from the audio of people speaking. Use a model to identify which language is being spoken. | This model is a fine-tuned version of openai/whisper-medium on the FLEURS subset of the google/xtreme_s dataset. It is used for language identification in audio classification tasks. |
| Code Retrieval COIR | What is the total volume of timber sold by each salesperson, sorted by salesperson? | ```sql CREATE TABLE salesperson (salesperson_id INT, name TEXT, region TEXT); INSERT INTO salesperson (salesperson_id, name, region) VALUES (1, 'John Doe', 'North'), (2, 'Jane Smith', 'South'); CREATE TABLE timber_sales (sales_id INT, salesperson_id INT, volume REAL, sale_date DATE); INSERT INTO timber_sales (sales_id, salesperson_id, volume, sale_date) VALUES (1, 1, 120, '2021-01-01'), (2, 1, 150, '2021-02-01'), (3, 2, 180, '2021-01-01'); ``` |
| Code Retrieval Design2Code |  |  |
| Code Retrieval Chart-Mimic |  |  |

| Task Name | | Query Example | Document Example |
|---|---|---|---|
| Reasoning BRIGHT | Intensive | Which part of the brain needs to be shut down to lose conciousness? Whether it is mechanical (trauma), chemical (anaesthesia) or electrical - which part of the brain is shut down to cause loss of consciousness? | There is no widely-accepted neurological structure that mediates 'consciousness.' Even if some structures have been shown to be necessary for consciousness, they have not been shown to be sufficient. This is true with anesthetic mechanisms as well — their ability to paralyze and block pain signals is fairly well-understood, but the mechanism of loss-of-consciousness is still unknown. 

 Still, 'consciousness' has to be there, somewhere between being awake and being dead, states which anesthetics can readily bridge (review): 

 Nevertheless, at some level of anesthesia between behavioral unresponsiveness and the induction of a flat EEG [indicating the cessation of the brain's electrical activity, one of the criteria for brain death (22)], consciousness must vanish. 

 Later in the same review: 

 The evidence from anesthesia and sleep states (Fig. 2–3) converges to suggest that loss of consciousness is associated with a breakdown of cortical connectivity and thus of integration, or with a collapse of the repertoire of cortical activity patterns and thus of information (Fig. 2). Why should this be the case? A recent theory suggests a principled reason: information and integration may be the very essence of consciousness (52). 

 This is consistent with my own take. Consciousness itself is the subjective experience of 'brain,' so it can't be lost, just poorly integrated. |
| Reasoning MRMR | Intensive | 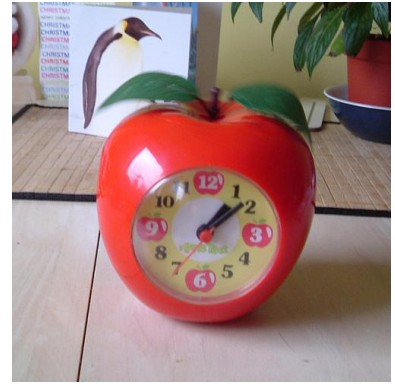 | This image includes clock, potted plant, Penguin but no chair. |
| Memory LongMemEval | Retrieval | I'm checking our previous chat about the shift rotation sheet for GM social media agents. Can you remind me what was the rotation for Admon on a Sunday? | can u create a shift rotation sheet for GM social media agents7 agents4 shifts as follows: "8 am till 4 pm - 12 pm till 8 pm - 4 pm till 12 am - 12 am till 8 am "2 days off for every agentsfor only a weekSunday - Saturdayagents names below:Admon Magdy Ehab Sara Mostafa Nemr Adam |

| Task Name | | Query Example | Document Example |
|---|---|---|---|
| Memory MMRC-R | Retrieval | What ancient civilization ruins were mentioned in this conversation? | User: I've been thinking lately about why ancient civilizations' ruins are so fascinating. How have they influenced our society today? 
 Assist: That's a really interesting question! Ancient ruins are not only witnesses to history but also reflect the wisdom and art of past civilizations. For example, Egypt's pyramids and Greece's Parthenon still influence our architecture and culture today. Do you have a particular interest in these ruins? 
 User: I'm especially interested in ancient Greek ruins. I also have some photos I've taken, and I'd love to know how the aesthetics of these ruins compare to modern architecture. 
 Assist: I'm glad you're sharing your photos! Ancient Greek architectural styles, especially the Doric, Ionic, and Corinthian columns, still influence our designs today. What kind of ruins are in your photos? We can compare them together. 
 User: Here's a photo I took during my recent trip to Greece. It seems like the lines in these ruins resonate a lot with modern architecture. What do you think? 
  
 Assist: This photo is really well taken! You're right—the simple lines of ancient Greek architecture do have a lot in common with modern minimalist design. Ancient Greek buildings focused on proportion and symmetry, and modern architecture often borrows from that, emphasizing both functionality and aesthetics. 
 descriptions |

*Table 9.* Detailed results of Deep QA datasets.

| Dataset → | Avg. | MuSiQue | BrowseComp-Plus | FreshStack-Angular | FreshStack-Godot | FreshStack-LangChain | FreshStack-Laravel | FreshStack-Yolo | DUDE | MVRB-CSR-Knowledge | MVRB-CSR-News-to-wiki | MVRB-CSR-Product | MVRB-CSR-Wiki-to-product | MVRB-SR-Chart | MVRB-SR-Doc | MVRB-SR-News | MVRB-SR-Paper | MVRB-SR-Product | MVRB-SR-Repo | MVRB-SR-Slide |
|---|---|---|---|---|---|---|---|---|---|---|---|---|---|---|---|---|---|---|---|---|
| bge-large | 17.22 | 29.64 | 6.75 | 16.03 | 12.40 | 17.55 | 20.46 | 17.70 | - | - | - | - | - | - | - | - | - | - | - | - |
| bge-m3 | 16.80 | 30.00 | 5.94 | 13.91 | 16.56 | 16.29 | 20.11 | 14.77 | - | - | - | - | - | - | - | - | - | - | - | - |
| mE5 | 16.33 | 30.46 | 7.40 | 14.39 | 10.69 | 16.50 | 19.02 | 15.87 | - | - | - | - | - | - | - | - | - | - | - | - |
| mGTE | 17.80 | 32.44 | 8.27 | 17.59 | 14.75 | 15.82 | 22.56 | 13.15 | - | - | - | - | - | - | - | - | - | - | - | - |
| Nomic-V2 | 15.79 | 31.95 | 5.69 | 10.24 | 11.31 | 16.96 | 18.98 | 15.37 | - | - | - | - | - | - | - | - | - | - | - | - |
| e5-mistral | 19.56 | 30.91 | 8.72 | 19.25 | 18.25 | 23.10 | 19.76 | 16.94 | - | - | - | - | - | - | - | - | - | - | - | - |
| NV-Embed-v2 | 17.63 | 34.75 | 16.41 | 16.12 | 13.47 | 13.69 | 17.15 | 11.85 | - | - | - | - | - | - | - | - | - | - | - | - |
| Qwen3$_{8B}$ | 27.29 | 33.91 | 20.09 | 27.85 | 25.82 | 22.71 | 32.40 | 28.28 | - | - | - | - | - | - | - | - | - | - | - | - |
| VISTA | 15.03 | 29.13 | 4.23 | 17.04 | 11.87 | 14.11 | 20.43 | 10.77 | 4.04 | 16.52 | 13.70 | 52.18 | 18.24 | 1.12 | 3.91 | 9.79 | 1.22 | 25.40 | 5.00 | 26.83 |
| VLM2Vec | 30.59 | 12.04 | 2.30 | 4.78 | 3.57 | 1.84 | 2.56 | 4.62 | 27.41 | 69.17 | 75.48 | 58.97 | 64.62 | 24.46 | 23.43 | 40.99 | 33.20 | 46.04 | 35.90 | 49.75 |
| gme-2B | 41.70 | 32.51 | 12.08 | 13.10 | 7.10 | 11.92 | 16.14 | 4.95 | 55.19 | 61.92 | 71.08 | 74.50 | 93.88 | 11.68 | 53.38 | 72.41 | 88.13 | 40.13 | 47.85 | 24.31 |
| gme-7B | 56.36 | 33.82 | 11.14 | 14.62 | 4.39 | 6.81 | 16.54 | 4.77 | 68.22 | 74.99 | 82.88 | 80.28 | 95.38 | 59.77 | 72.40 | 83.97 | 95.09 | 88.75 | 88.17 | 88.84 |
| mmE5 | 47.72 | 20.22 | 10.89 | 16.83 | 6.90 | 17.39 | 18.95 | 9.08 | 38.02 | 77.95 | 82.74 | 73.51 | 87.56 | 46.92 | 41.20 | 69.29 | 70.75 | 75.57 | 71.91 | 71.03 |
| Jina-v4 | 52.78 | 30.16 | 4.92 | 17.35 | 17.89 | 16.61 | 20.10 | 14.24 | 61.20 | 53.86 | 50.39 | 72.93 | 88.03 | 64.76 | 58.35 | 79.23 | 92.15 | 88.37 | 84.38 | 87.86 |
| Qwen3VL$_{2B}$ | 59.16 | 31.71 | 8.43 | 23.46 | 24.49 | 21.52 | 25.56 | 22.85 | 66.80 | 83.13 | 85.75 | 81.86 | 93.06 | 52.75 | 62.73 | 81.17 | 92.15 | 89.77 | 88.86 | 87.97 |
| VLM2Vec-v2 | 47.29 | 26.32 | 7.96 | 11.07 | 8.47 | 8.10 | 11.52 | 9.63 | 54.18 | 79.41 | 78.79 | 65.55 | 84.73 | 50.97 | 48.78 | 51.30 | 69.31 | 75.20 | 71.69 | 85.52 |

*Table 10.* Detailed results of Code and Tool datasets.

| Dataset → / Model ↓ | Avg. | ToolRet-Code | ToolRet-Customized | ToolRet-Web | ReFocus | Avg. | CoIR-APPS | CoIR-CodeFeedback-MT | CoIR-CodeFeedback-ST | CoIR-CodeTransOcean-Contest | CoIR-CodeTransOcean-DL | CoIR-CosQA | CoIR-StackOverflowQA | CoIR-SyntheticText2SQL | Design2Code | ChartMimic |
|---|---|---|---|---|---|---|---|---|---|---|---|---|---|---|---|---|
| bge-large | 47.06 | 52.06 | 46.50 | 42.63 | - | 44.94 | 6.70 | 36.73 | 72.10 | 54.34 | 21.63 | 33.75 | 83.36 | 50.90 | - | - |
| bge-m3 | 44.27 | 51.23 | 42.06 | 39.52 | - | 47.43 | 14.75 | 47.85 | 69.29 | 62.59 | 29.29 | 28.52 | 80.54 | 46.64 | - | - |
| mE5 | 42.04 | 45.94 | 42.21 | 37.96 | - | 52.27 | 28.48 | 35.90 | 72.36 | 75.65 | 32.02 | 31.55 | 90.37 | 51.56 | - | - |
| mGTE | 42.63 | 50.28 | 41.00 | 36.62 | - | 52.01 | 11.96 | 52.24 | 77.33 | 67.73 | 35.69 | 33.68 | 87.10 | 50.35 | - | - |
| Nomic-V2 | 46.72 | 53.88 | 44.28 | 42.00 | - | 40.54 | 5.64 | 28.50 | 63.56 | 43.19 | 23.41 | 28.10 | 74.35 | 57.55 | - | - |
| e5-mistral | 46.12 | 55.40 | 42.97 | 39.99 | - | 58.75 | 26.57 | 58.26 | 75.81 | 88.70 | 35.17 | 28.55 | 93.06 | 63.88 | - | - |
| NV-Embed-v2 | 48.55 | 57.30 | 48.41 | 39.96 | - | 47.37 | 25.41 | 62.58 | 68.52 | 63.88 | 26.00 | 31.64 | 35.23 | 65.72 | - | - |
| Qwen3$_{8B}$ | 37.27 | 35.88 | 43.87 | 32.06 | - | 57.00 | 30.56 | 80.69 | 83.06 | 89.80 | 10.79 | 39.78 | 53.87 | 67.46 | - | - |
| VISTA | 35.37 | 49.76 | 42.42 | 40.10 | 9.19 | 38.21 | 5.72 | 33.79 | 69.80 | 45.77 | 22.23 | 32.48 | 79.93 | 54.54 | 7.90 | 29.89 |
| VLM2Vec | 17.98 | 20.25 | 31.14 | 15.64 | 4.91 | 26.19 | 3.73 | 33.02 | 32.05 | 27.12 | 33.90 | 9.19 | 25.71 | 37.53 | 25.00 | 34.61 |
| gme-2B | 42.56 | 57.60 | 47.84 | 43.98 | 20.81 | 54.59 | 24.43 | 75.66 | 80.28 | 85.13 | 34.30 | 32.20 | 85.59 | 57.08 | 30.07 | 41.14 |
| gme-7B | 45.09 | 61.56 | 49.03 | 46.83 | 22.97 | 64.35 | 48.02 | 85.73 | 83.21 | 90.56 | 34.01 | 32.26 | 83.97 | 60.21 | 31.68 | 93.82 |
| mmE5 | 30.60 | 47.03 | 39.87 | 26.64 | 8.87 | 55.42 | 16.32 | 71.09 | 64.87 | 75.15 | 35.09 | 22.92 | 75.03 | 56.46 | 45.15 | 92.14 |
| Jina-v4 | 36.07 | 55.12 | 45.44 | 41.17 | 2.55 | 70.33 | 76.53 | 69.87 | 84.46 | 91.43 | 27.65 | 29.20 | 93.02 | 71.19 | 62.04 | 97.91 |
| Qwen3VL$_{2B}$ | 42.63 | 52.34 | 46.85 | 49.38 | 21.97 | 70.46 | 60.91 | 85.33 | 83.97 | 88.70 | 35.92 | 35.12 | 90.55 | 65.16 | 60.51 | 98.43 |
| VLM2Vec-v2 | 24.58 | 29.41 | 33.03 | 31.25 | 4.63 | 53.04 | 14.03 | 75.62 | 70.44 | 73.58 | 35.03 | 11.74 | 63.32 | 65.54 | 23.70 | 97.40 |

*Table 11.* Detailed results of reasoning-intensive retrieval datasets.

| Dataset → | Avg. | BRIGHT-aops | BRIGHT-biology | BRIGHT-earth-science | BRIGHT-economics | BRIGHT-psychology | BRIGHT-robotics | BRIGHT-stackoverflow | BRIGHT-sustainable-living | BRIGHT-theoremqa-questions | BRIGHT-theoremqa-theorems | RAR-B-TempReasonL1 | RAR-B-TempReasonL2Context | RAR-B-TempReasonL2Fact | RAR-B-TempReasonL2Pure | RAR-B-TempReasonL3Context | RAR-B-TempReasonL3Fact | RAR-B-TempReasonL3Pure | MRMR-Theorem | MRMR-Design | MRMR-Traffic | MRMR-Negation |
|---|---|---|---|---|---|---|---|---|---|---|---|---|---|---|---|---|---|---|---|---|---|---|
| bge-large | 16.29 | 0.94 | 26.11 | 39.63 | 19.06 | 22.59 | 10.33 | 13.99 | 20.51 | 13.61 | 22.50 | 1.46 | 17.72 | 24.25 | 2.35 | 14.65 | 20.62 | 6.67 | - | - | - | - |
| bge-m3 | 18.95 | 1.51 | 27.31 | 37.97 | 20.67 | 31.61 | 16.01 | 19.17 | 17.91 | 14.30 | 24.02 | 0.99 | 22.32 | 33.20 | 0.67 | 19.24 | 30.04 | 5.28 | - | - | - | - |
| mE5 | 19.08 | 1.93 | 17.87 | 33.95 | 15.39 | 18.50 | 14.96 | 16.70 | 15.59 | 14.45 | 23.36 | 1.22 | 37.63 | 35.37 | 2.95 | 32.60 | 31.90 | 10.01 | - | - | - | - |
| mGTE | 13.56 | 1.40 | 20.50 | 35.52 | 17.03 | 23.91 | 12.51 | 12.84 | 16.58 | 13.21 | 20.45 | 1.08 | 10.25 | 15.07 | 1.51 | 9.70 | 13.81 | 5.13 | - | - | - | - |
| Nomic-V2 | 17.12 | 0.65 | 29.66 | 35.38 | 21.76 | 27.62 | 11.46 | 17.04 | 19.07 | 15.07 | 22.62 | 0.96 | 17.89 | 26.80 | 0.28 | 15.48 | 22.16 | 7.13 | - | - | - | - |
| e5-mistral | 18.07 | 1.91 | 26.90 | 39.24 | 20.45 | 22.43 | 11.59 | 13.86 | 17.42 | 13.56 | 26.69 | 1.23 | 21.89 | 29.61 | 5.71 | 18.77 | 24.66 | 11.23 | - | - | - | - |
| NV-Embed-v2 | 13.46 | 1.63 | 18.88 | 2.58 | 10.37 | 8.35 | 9.23 | 4.95 | 11.26 | 18.80 | 21.74 | 2.95 | 20.11 | 34.77 | 6.74 | 16.35 | 27.91 | 12.14 | - | - | - | - |
| Qwen3$_{8B}$ | 15.62 | 2.15 | 19.48 | 10.45 | 16.86 | 20.92 | 8.33 | 2.00 | 21.94 | 19.41 | 26.96 | 1.22 | 19.96 | 31.29 | 6.81 | 17.23 | 28.01 | 12.48 | - | - | - | - |
| VISTA | 15.05 | 0.85 | 26.34 | 35.45 | 18.82 | 22.27 | 9.29 | 14.53 | 18.83 | 14.45 | 22.29 | 1.07 | 9.92 | 17.24 | 1.29 | 8.17 | 13.36 | 5.21 | 19.93 | 24.50 | 21.53 | 10.71 |
| VLM2Vec-lora | 11.34 | 1.96 | 14.78 | 10.28 | 2.10 | 3.96 | 0.76 | 0.78 | 8.55 | 17.57 | 8.32 | 1.28 | 11.59 | 31.64 | 5.48 | 8.08 | 29.28 | 9.22 | 18.23 | 24.25 | 20.06 | 10.05 |
| gme-2B | 19.59 | 1.21 | 27.70 | 32.96 | 20.19 | 28.11 | 11.89 | 15.53 | 16.91 | 15.61 | 25.41 | 0.83 | 15.19 | 27.72 | 2.22 | 12.84 | 21.83 | 6.14 | 26.45 | 53.43 | 37.54 | 11.69 |
| gme-7B | 20.21 | 0.85 | 25.23 | 29.96 | 15.01 | 27.10 | 10.35 | 11.91 | 18.39 | 15.86 | 26.49 | 1.23 | 23.06 | 36.07 | 4.49 | 17.71 | 28.75 | 11.81 | 29.83 | 43.72 | 36.54 | 11.07 |
| mmE5 | 18.33 | 1.18 | 31.78 | 29.07 | 13.77 | 22.98 | 8.26 | 7.30 | 14.97 | 11.73 | 22.06 | 2.14 | 26.76 | 36.66 | 8.88 | 19.67 | 30.29 | 13.78 | 16.64 | 22.65 | 32.17 | 12.25 |
| Jina-v4 | 17.71 | 1.14 | 26.55 | 27.95 | 17.63 | 21.77 | 9.74 | 12.80 | 13.42 | 14.51 | 21.86 | 1.56 | 16.59 | 28.35 | 2.95 | 12.81 | 24.07 | 8.00 | 21.38 | 50.54 | 25.14 | 13.12 |
| Qwen3VL$_{2B}$ | 22.55 | 2.40 | 30.24 | 39.97 | 25.73 | 32.65 | 17.12 | 17.64 | 21.71 | 14.62 | 26.35 | 1.28 | 14.87 | 29.36 | 5.11 | 11.99 | 22.19 | 8.17 | 33.95 | 61.24 | 44.57 | 12.42 |
| VLM2Vec-v2 | 14.01 | 4.93 | 18.42 | 16.53 | 6.35 | 11.34 | 4.89 | 2.32 | 6.33 | 17.86 | 5.43 | 1.04 | 12.20 | 25.96 | 2.39 | 10.64 | 25.48 | 7.10 | 21.54 | 46.02 | 34.98 | 12.52 |

*Table 12.* Detailed results of memory retrieval datasets.

| Dataset → | Avg. | LoCoMo-adversarial | LoCoMo-multi-hop | LoCoMo-open-domain | LoCoMo-single-hop | LoCoMo-temporal | LongMemEval | SRA-brief | SRA-full | MMRC-R-round | MMRC-R-session |
|---|---|---|---|---|---|---|---|---|---|---|---|
| bge-large | 49.90 | 41.74 | 40.29 | 34.06 | 58.03 | 62.09 | 72.98 | 40.10 | 49.94 | - | - |
| bge-m3 | 48.37 | 33.69 | 38.45 | 29.24 | 58.14 | 62.99 | 75.90 | 39.87 | 48.72 | - | - |
| mE5 | 50.03 | 41.51 | 40.08 | 30.81 | 63.79 | 65.71 | 68.85 | 38.41 | 51.05 | - | - |
| mGTE | 38.99 | 19.34 | 24.73 | 25.71 | 44.78 | 51.71 | 47.12 | 44.81 | 53.72 | - | - |
| Nomic-V2 | 45.25 | 33.51 | 35.98 | 25.42 | 52.62 | 59.09 | 71.16 | 37.51 | 46.69 | - | - |
| e5-mistral | 56.62 | 52.97 | 40.25 | 31.91 | 64.98 | 63.54 | 79.73 | 57.53 | 62.03 | - | - |
| NV-Embed-v2 | 59.18 | 57.91 | 43.06 | 32.34 | 70.13 | 67.20 | 70.78 | 61.98 | 70.02 | - | - |
| Qwen3$_{8B}$ | 57.11 | 40.15 | 41.61 | 34.73 | 61.02 | 64.73 | 73.60 | 67.50 | 73.52 | - | - |
| VISTA | 49.24 | 34.47 | 30.42 | 23.85 | 51.05 | 53.78 | 71.23 | 34.67 | 45.49 | 73.30 | 74.10 |
| VLM2Vec | 44.51 | 36.46 | 25.43 | 18.41 | 44.84 | 38.45 | 60.38 | 41.83 | 41.02 | 69.39 | 68.91 |
| gme-2B | 52.90 | 25.82 | 24.46 | 23.37 | 44.67 | 48.73 | 76.97 | 55.07 | 59.07 | 87.34 | 83.50 |
| gme-7B | 57.23 | 33.60 | 27.79 | 24.64 | 53.93 | 48.45 | 77.05 | 61.80 | 73.73 | 87.86 | 83.51 |
| mmE5 | 47.75 | 34.69 | 25.97 | 21.16 | 48.24 | 57.30 | 65.18 | 49.69 | 45.39 | 56.54 | 73.37 |
| Jina-v4 | 48.42 | 17.23 | 22.90 | 21.83 | 41.65 | 49.82 | 65.84 | 41.94 | 58.91 | 84.83 | 79.26 |
| Qwen3VL$_{2B}$ | 60.32 | 46.15 | 39.72 | 29.64 | 56.88 | 66.21 | 71.44 | 61.97 | 62.28 | 83.91 | 85.00 |
| VLM2Vec-v2 | 51.39 | 30.30 | 24.52 | 20.37 | 48.65 | 52.09 | 74.77 | 48.37 | 59.56 | 75.25 | 80.03 |

