# OpenReview forum: "AOEB: Benchmarking Agent-Oriented Multimodal Embeddings"
_ICML.cc/2026/Conference — ICML 2026 regular_

### Official Review · Reviewer_LN15 · 2026-03-12

**Soundness:** 3
**Presentation:** 3
**Significance:** 2
**Originality:** 2
**Overall Recommendation:** 4
**Confidence:** 4

**Summary:**

This paper introduces the Agent-Oriented Embedding enchmark (AOEB), a new evaluation suite for embeddin models specifically designed for agentic LLM applictions. The benchmark addresses a perceived gap betwen existing benchmarks (e.g., MTEB, BEIR) that focus n general-purpose retrieval and the specialized need of LLM agents. AOEB covers five meta-tasks: Deep QA Tool Retrieval, Code Retrieval, Reasoning-intensiveRetrieval, and Memory Retrieval, each with both textand image modalities, yielding 66 sub-tasks drawn frm 21 datasets. The authors also introduce AOEB-T, a ext-only subset. The paper evaluates a range of multmodal and text-only embedding models and proposes usng Borda count ranking for a more robust holistic coparison across tasks. Downstream experiments on RAG asks (FreshStack and BrowseCompPlus) confirm that beter AOEB scores correlate with improved end-to-end aent performance.

**Compliance With Llm Reviewing Policy:**

Affirmed.

**Final Justification:**

The authors rebuttal has fully resolved my concerns with solid experimental results, so I am raising my score to 4.

**Key Questions For Authors:**

1. Could the authors clarify what portion of AOEB is newly constructed versus reformulated or directly reused from existing datasets? A clearer breakdown would help assess the benchmark’s technical novelty.

2. How were the compared baselines selected? In particular, were newer models such as VLM2Vec-V2, Qwen3-VL-Embedding, and NVIDIA Omni-Embed Nemotron omitted due to submission timing, compute budget, or implementation constraints?

3. How sensitive are the results to the interleaved-input adaptation method? For interleaved image-text datasets, how much do model rankings change between Merge-image and Visual-doc?

4. Did the authors evaluate Recall@k and choose not to report it? Since many agent pipelines depend on retrieving the correct item at all, recall seems like an important complementary metric.

5. Could the authors clarify the paper’s novelty relative to prior multimodal trajectory retrieval work, such as Zhang et al. (2025), Universal Retrieval for Multimodal Trajectory Modeling? It would be helpful to distinguish the novelty of the unified benchmark from the broader problem setting.

6. Could the authors revise Figure 5 for readability? The legend in the upper subfigure overlaps the plot area, and the x-axis labels in the lower subfigure are too tightly spaced.

**Limitations:**

Yes.

**Strengths And Weaknesses:**

Strengths:
1. Comprehensive task coverage: AOEB spans five meta-tasks, Deep QA, Tool Retrieval, Code Retrieval, Reasoning-intensive Retrieval, and Memory Retrieval, with 66 subtasks across 21 datasets, providing broad coverage of agent-relevant scenarios. The inclusion of both text and image modalities is a notable contribution compared with prior text-only benchmarks.

2. Practical downstream validation: The paper includes end-to-end agent evaluation on FreshStack RAG and BrowseComp-Plus, confirming that AOEB scores correlate with real-world performance. This strengthens the benchmark’s practical relevance and distinguishes it from purely metric-driven comparisons.

Weaknesses:
1. Limited novelty in benchmark construction: Most of AOEB appears to be assembled from existing datasets and benchmarks, with limited newly created data or annotations. As a result, the main contribution is more in benchmark integration and task organization than in new dataset creation or modeling advances.

2. Incomplete and somewhat outdated baseline selection: The evaluated model set does not fully reflect the current multimodal retrieval landscape. In particular, the paper includes VLM2Vec-LoRA but omits stronger or newer relevant baselines such as VLM2Vec-V2, Qwen3-VL-Embedding, NVIDIA Omni-Embed Nemotron, and, under the visual-document setting, ColQwen2.

3. Interleaved-input handling is heuristic and may distort evaluation: The proposed Merge-image and Visual-doc strategies are practical heuristics, but they are not fully principled. In particular, Merge-image may sacrifice image resolution and remove fine-grained visual details, which could materially affect performance on multimodal retrieval tasks.

4. Evaluation metrics are narrow: The paper mainly reports nDCG@10, along with aggregate summaries such as micro average, type average, and Borda rank, but does not report Recall@k. For agent-oriented retrieval, recall is important because downstream failure often occurs when the correct item is not retrieved at all.

---

> ### Author Rebuttal · Authors · 2026-03-31
>
> We sincerely thank your for the valuable questions, which we respond accordingly. We believe these improvements will strengthen our work. We welcome any further discussion and hope our responses address your concerns. If so, we  would greatly appreciate it if you could consider raising your score.
>
> ---
>
> ### Q1
> > Limited novelty
>
> We agree that, from a technical perspective, the amount of newly created data in AOEB is limited. Among the 17 datasets in AOEB, 5 are reformulated (DUDE, ReFocus, Design2Code, ChartMimic, and MLDR), and 1 with new annotations (MMRC‑R, generate queries), while the remaining datasets are directly reused (but filtered out some subset).
> We will actively expand AOEB with newly constructed datasets in future iterations, including tasks such as multimodal trajectory retrieval (mentioned by the reviewer, very insightful), skill retrieval, and others.
>
> Our primary goal in this work is to introduce a more systematic benchmark focused on agent‑oriented retrieval, integrating a set of previously isolated subtasks and providing the community with a new perspective on retrieval capabilities for agentic workflows.
>
> ### Q2
> > Outdated multimodal models
>
> Thank you for the suggestion. We evaluate VLM2Vec‑V2 and Qwen3‑VL‑Embedding results as suggested (please refer to rebuttal to #GMjy and the following Q4), and will continue expanding the set of evaluated models.
>
> When selecting baselines, our principle was to include models that had publicly accepted papers by late 2025, which is a relatively conservative strategy. We will include more models.
>
> ### Q3
> > Interleaved-input
>
> We agree they are heuristic approaches, adopted due to the lack of native multi-image support in current open-source models. Sensitivity to this adaptation depends on model specifics: models trained on visual document retrieval benefit from Visual-doc, whereas earlier VISTA models (without such training) perform worse than with Merge-image.
>
> The table below compares results (recall@10) on two representative subsets. **rank(mi-vd)** indicates the rank difference (rank_in_merge_image - rank_in_visual_doc).
> **traffic** / **traffic-vd** denote the respective scores with *Merge-image* or *Visual-doc*. Results show that model rankings are largely stable between settings. Only the VISTA model shows a significant ranking drop (-3) in Visual-doc mode , consistent with its lack of visual document training.
>
> |model|rank(mi-vd)-traffic|rank(mi-vd)-theorem|traffic|traffic-vd|diff(vd-mi)|theorem|theorem-vd|diff(vd-mi)|
> |-|-|-|-|-|-|-|-|-|
> |GME-2B|0|1|46.41|51.56|5.15|32.50|33.22|0.73|
> |GME-7B|0|-1|52.13|55.01|2.89|37.86|29.11|-8.75|
> |Jina-v4|1|0|23.54|38.91|15.37|28.66|28.63|-0.03|
> |mmE5|0|2|30.21|45.97|15.76|8.10|22.64|14.54|
> |Qwen3VL-2B|0|0|57.49|61.56|4.08|43.26|37.65|-5.61|
> |VISTA|-3|-3|28.85|12.57|-16.28|25.44|1.01|-24.43|
> |VLM2Vec|1|0|12.29|36.41|24.12|15.73|17.58|1.85|
> |VLM2Vec2|1|1|5.63|16.56|10.94|2.99|1.30|-1.69|
>
> ### Q4
> > Recall scores
>
> We agree on the importance of recall. We computed recall scores in our evaluation; a brief table is shown below.
> As table shown, the performance ranking trends across models under recall scores are consistent with those observed under NDCG scores (rebuttal to #GMjy).
>
> **Recall socres:**
> |Model|Context|Size|Rank_Borda|Avg.Type|Avg.66|DeepQA19|Tool4|Code10|Reasoning21|Memory12|
> |-|-|-|-|-|-|-|-|-|-|-|
> |VISTA|512|0.2B|6|39.33|31.99|18.85|41.52|52.97|20.19|63.14|
> |VLM2Vec|8192|4B|7|29.61|26.98|37.25|23.99|23.42|9.84|53.56|
> |GME-2B|8192|2B|4|53.84|49.45|58.64|/50.97/|73.02|25.04|61.54|
> |GME-7B|8192|8B|2|/56.29/|/51.79/|59.68|**52.80**|77.47|/27.62/|/63.88/|
> |mmE5|8192|11B|5|40.88|35.58|29.96|40.78|68.46|20.57|44.62|
> |Jina-v4|8192|3B|3|55.55|50.92|/60.71/|48.03|**85.96**|22.09|60.97|
> |Qwen3VL-2B|8192|2B|1|**59.02**|**54.73**|**62.98**|50.23|/85.92/|**28.37**|**67.60**|
> |VLM2Vec2|8192|2B|8|21.98|19.66|18.82|12.80|39.93|8.00|30.35|
>
> ### Q5
> > multimodal trajectory retrieval work
>
> Thank you for raising this point. This is a highly relevant and forward‑looking work. It highlights the role of embedding models in agent‑related tasks and introduces a very novel trajectory retrieval task to support GUI agent execution. This represents an important early contribution within the broader space of agent‑oriented retrieval tasks and provides a unique and valuable task.
> We believe that incorporating trajectory retrieval within AOEB would make the benchmark even more comprehensive. *We sincerely appreciate the reviewer for pointing out this work, it is indeed very helpful, and we will make sure to cite and highlight it, as well as discuss it in detail in our revision*. We are also very sorry and regretful that we overlooked this relevant work during our literature review.
>
> ### Q6
> > revise Figure 5
>
> Thank you for pointing this out. Sorry this is an incorrect version of Figure 5 by mistake. We will fix the issues to improve readability in the revised version.

---

> > ### Author Rebuttal · Reviewer_LN15 · 2026-04-03
> >
> > I am satisfied with most of the authors' responses except Q3. Any retrieval model based on the Qwen series (from Qwen2-VL onward, released by Aug 2024) already supports native multi-image interleaved input. Merging images, which downsamples them, inevitably sacrifices retrieval performance, particularly in agent tasks, making those experimental results unreliable. I consider this a severe limitation that cannot be addressed within this rebuttal.
> > That said, I am willing to raise my score to 3, in recognition of the authors' effort in providing additional comparable baselines with recall metrics and their efforts to reposition the paper with other related work.
> >
> > Separately, I strongly disagree with Reviewer CXkY's perspective that other reviewers are requiring experiments with models released close to the submission deadline. VLM2Vec-V2 was released by May 2025 and NVIDIA Omni-Embed Nemotron by October 2025, both at least three months before the deadline. I would not insist on Qwen3-VL-Embedding results given its proximity to the deadline, but expecting coverage of established models is reasonable. Given the growing number of retrieval benchmarks, a strong benchmark contribution should be as comprehensive as possible and clearly motivate why this additional benchmark is needed.

---

> > > ### Author Response · Authors · 2026-04-06
> > >
> > > Thank you very much for the acknowledgement and willing to raise your score.
> > >
> > > We would like to clarify and provide additional evidence for Q3.
> > >
> > > **Clarification**. We agree that MLLM *backbone models* from the Qwen2-VL series support native interleaved inputs. Our original statement was imprecise. Our point is that existing open-source *embedding models* (e.g., Qwen3-VL-Embedding, VLM2Vec, GME) do not explicitly support or evaluate interleaved multi-image inputs in their model card standard usage or training setup from the paper.
> > >
> > > Hence, in our experiments, following recent studies on multimodal interleaved retrieval [1,2], we adopted two simple yet effective adaptation strategies. To ensure robustness, we reported the maximum performance achieved by either method for all models.
> > >
> > > **Additional experiments**. To directly address your concern, we modify the inference pipeline of Qwen3-VL-Embedding and VLM2Vec-v2 (Qwen2-VL backbone) to enable native interleaved multi-image input, and compared it with the merge-image (mi)  and vis-doc (vd) strategies.
> > >
> > > | Model      | traffic-mi | traffic-interleave | traffic-vd | theorem-mi | theorem-interleave | theorem-vd |
> > > |------------|------------|------------------|------------|------------|------------------|------------|
> > > | Qwen3-VL-2B |     44.57  |           44.84  |     44.30  |     33.95  |           33.93  |     30.10  |
> > > | VLM2Vec-V2     |     23.87  |           23.88  |     34.98  |     21.54  |           21.32  |     17.77  |
> > >
> > > We observe that, the performance difference is consistently small.
> > > These results suggest that, despite potential information loss from image merging, the overall retrieval performance remains stable, likely because current embedding models are not specifically optimized for interleaved multi-image reasoning.
> > >
> > > Meanwhile, the two strategies, merge-image and vis-doc, exhibit complementary strengths across different subsets. Therefore, to ensure an accurate assessment of performance, we reported the maximum performance achieved by either method for all models in our main evaluation. Consequently, we believe that our current evaluation protocol provides a reasonable and reliable approximation for interleaved datasets.
> > >
> > > We hope this clarification and the additional experimental results could address your concern. Thank you for your time!
> > >
> > > ---
> > >
> > > [1] MRMR: A Realistic and Expert-Level Multidisciplinary Benchmark for Reasoning-Intensive Multimodal Retrieval. ICLR 2026.
> > >
> > > [2] Towards Text-Image Interleaved Retrieval. ACL 2025.

---

### Official Review · Reviewer_CXkY · 2026-03-12

**Soundness:** 3
**Presentation:** 3
**Significance:** 3
**Originality:** 3
**Overall Recommendation:** 5
**Confidence:** 4

**Summary:**

Review

This paper  introduces a new benchmark to evaluating embeddings for agentic settings. This address and existing gap in the field that current benchmarks only partially addresses.
The benchmark build on the well-established MTEB framework hereby making their work easily accessible and ensuring future compatibility.

Generally the work is well-written. I would have loved to have seen the content merged into MTEB as PRs to public repositories are not always trivial and might reveal data or implementation issues that might impact the result. I do however understand that this impact anonymity, so under the conditions I believe it is an appropriate choice.

**Compliance With Llm Reviewing Policy:**

Affirmed.

**Final Justification:**

I believe the authors have appropriately addressed my comments and provided a meaningful contribution to the field. I would be happy to see it accepted and integrated into MTEB.

**Key Questions For Authors:**

Questions:

- Can you add a limitations section?
- Can you expand on you rationality for not including multiple languages?
- How long does it take to run the full benchmark?
- Can you provide some sort of correlation matrix of the tasks to ensure that they are in fact diverse?

Suggestion

- In figure 2 you write higher is better, but I suspect you mean lower is better (1 is better than 2). I would probably rephrase to avoid the ambiguity

- I understand that you want to use shorter names, but I would provide exact model ids (ideally with the revision as well) in the appendix

- presnet -> present (~l212)

- The authors might be overly critical of general benchmarks, which, though I agree not use an LLM). I would tone it down without changing the conclusions.


Notes

- I do believe the MTEB support mixtures of multiple modality settings introduced in MIEB with images and expanded by MAEB to also include audio. Though it is possible that this work was done before then.that this benchmark is needed, still represent a lare portion of search use-cases (many of which do

**Limitations:**

I did not find a limitation section in the paper. I believe the paper would benefit from a limitations section.

**Strengths And Weaknesses:**

Strengths

- Addresses the current lack of a benchmark for emerging applications of embeddings

- While code and data are not currently shared, they state that they will contribute them to MTEB, ensuring integration and maintenance within a larger framework.

- Task design and curation are reasonable, though I believe many of the trends of RTEB stem from MMTEB

- They evaluate the downstream implications of their benchmark on an agentic use-case and see that it transfers


Weaknesses:

- Code and data are not currently shared, but see strengths.

- Keeping the datasets sufficiently small is mentioned as a priority, but I do not see anything on measuring runtime or in other ways ensuring that the benchmark is accessible.

- Multilinguality is not addressed

---

> ### Author Rebuttal · Authors · 2026-03-31
>
> We thank the reviewer for the positive feedback and insightful questions. We have provided detailed responses above and believe they help further clarify and strengthen the paper. We hope our responses address your concerns and are happy to provide additional clarification if needed.
>
> ---
>
> ### Q1
> > Code and Limitation
>
> We will release the full benchmark on GitHub and HuggingFace and incorporates revisions based on the insightful feedback from all reviewers. Currently we share the code of evaluation and data acquisition via https://anonymous.4open.science/r/aoeb-anonymous/ and will keep updating.
>
> We will add a discussion of limitations in the revision. In summary, the main limitations are threefold:
>
> 1. Limited model coverage. We currently evaluate a set of representative and well-documented models, but some of the latest models are not yet included. We will continuously expand the benchmark to include newer models in MMEB and MIEB.
> 2. Room for task expansion. The current benchmark can be further extended in several directions. First, multilinguality: most current tasks are in English, while modern embedding models are often multilingual. Second, task design: as agent-related scenarios evolve rapidly, new tasks such as agent trajectory retrieval and agent skill retrieval can be incorporated.
> 3. Handling overfitting. At present, AOEB mainly mitigates overfitting by introducing diverse and newly constructed tasks. However, once AOEB is publicly released, the risk of overfitting remains. This can be further addressed by continuously updating the data and maintaining a portion of private test sets.
>
>
> ### Q2
> > Benchmark Runtime
>
> For 2B-3B level models, with proper optimization (such as flash_attention and half-precision), it takes about 50 hours to run all tests on single RTX A6000 GPU. And for 7B-level models, it's about 190 hours. We will add the runtime details in revision.
>
> ### Q3
> > Multilinguality
>
> Thank you for pointing this out. As an early-stage work, current datasets are primarily English. In future iterations, we plan to translate or expand the benchmark to support additional languages.
> Most agent-related work is currently based on English data. Similar to the development patterns of other benchmarks, such as MTEB, CMTEB, and MMTEB, we believe that focusing on English first and then progressively expanding to other languages is a reasonable and practical trajectory.
>
> ### Q4
> > Dataset diversity
>
> We compute embeddings for all datasets (sampled datapoints and average their embeddings) and visualized the similarity matrix (see this [anonymous similarity.png](https://anonymous.4open.science/r/aoeb-anonymous/assets/similarity_heatmap.png)). The low inter-dataset similarities confirm reasonable diversity across tasks to some extent.
>
> ### Q5
> > Suggestions and Notes
>
> Thanks for your suggestions, we will make changes accordingly.
>
> 1. Typos. Yes, lower is better. Will correct it, also ~l212.
> 2. We list exact model ids here, and will add it to the appendix.
>    - VISTA (`BAAI/bge-visualized`)
>    - VLM2Vec (`TIGER-Lab/VLM2Vec-LoRA`)
>    - GME_2B (`Alibaba-NLP/gme-Qwen2-VL-2B-Instruct`)
>    - GME_7B (`Alibaba-NLP/gme-Qwen2-VL-7B-Instruct`)
>    - mmE5 (`intfloat/mmE5-mllama-11b-instruct`)
>    - Jina-v4 (`jinaai/jina-embeddings-v4`)
>    - BGE_Large (`BAAI/bge-large-en-v1.5`)
>    - BGE-m3 (`BAAI/bge-m3`)
>    - mE5 (`intfloat/multilingual-e5-large-instruct`)
>    - mGTE (`Alibaba-NLP/gte-multilingual-base`)
>    - Nomic-v2 (`nomic-ai/nomic-embed-text-v2-moe`)
>    - E5_Mistral (`intfloat/e5-mistral-7b-instruct`)
>    - NV-Embed-v2 (`nvidia/NV-Embed-v2`)
>    - Qwen3-Embed-8B (`Qwen/Qwen3-Embedding-8B`)
>    - Qwen3VL-Embed-2B (`Qwen/Qwen3-VL-Embedding-2B`)
>    - VLM2Vec2 (`VLM2Vec/VLM2Vec-V2.0`)
> 3. Overly critical of general benchmarks. Thank you for pointing this out, will carefully tune the language.
> 4. Regarding mixed-modality settings, we checked the latest MTEB code. It supports mixing modalities within a datapoint (e.g., `text,image`), but currently enforces corpus-level uniformity where all documents share the same modality combination. But the corpus-mixed setting requires handling heterogeneous documents (`text`, `text,image`, `image`) within the same corpus. To support this in MTEB, we think grouping data by modality type in `_prepare_dataset`, encoding groups separately, and concatenating embeddings for unified search would be a feasible case.  We would greatly appreciate any corrections if we have misunderstood anything.
>
> Perhaps make changes in  https://github.com/embeddings-benchmark/mteb/blob/main/mteb/_create_dataloaders.py

---

> > ### Author Rebuttal · Reviewer_CXkY · 2026-04-01
> >
> > The authors address most concerns that I had. I think the given rating score is still appropriate.
> >
> > I must also say that I think the other reviewers missed the work that is present in developing a benchmark as well as missed the benefits of having the benchmark integrated in an existing framework, which provides both reproducibility, longevity, version and more. Similarly, they also require models that were released close to the submission deadline, and while I agree that these should be included in the public benchmark or even in the final paper, I don't think it is a major weakness.

---

> > > ### Author Response · Authors · 2026-04-06
> > >
> > > Thank you very much for the acknowledgement.

---

### Official Review · Reviewer_GMjy · 2026-03-13

**Soundness:** 2
**Presentation:** 3
**Significance:** 2
**Originality:** 2
**Overall Recommendation:** 3
**Confidence:** 5

**Summary:**

This paper introduces AOEB, a new benchmark designed to evaluate embedding models in agent-oriented retrieval scenarios. The authors argue that existing embedding benchmarks focus primarily on general-purpose retrieval tasks and therefore fail to reflect the requirements of emerging LLM-based agent systems.

The benchmark consists of 66 subtasks across textual and visual modalities, aggregated from existing datasets and newly curated ones. The authors evaluate a wide range of embedding models (both text-only and multimodal). Experimental results suggest that models performing well on standard benchmarks do not necessarily perform well on AOEB, indicating a gap between existing evaluation paradigms and agent-oriented applications.

**Compliance With Llm Reviewing Policy:**

Affirmed.

**Key Questions For Authors:**

Please include stronger and more recent models in the evaluation.

**Limitations:**

yes

**Strengths And Weaknesses:**

**Well-motivated benchmark design:** The proposed benchmark systematically organizes evaluation around five agent-oriented retrieval capabilities (deep QA, code retrieval, tool retrieval, reasoning retrieval, and memory retrieval), which closely align with core functionalities required by LLM agents.

**Comprehensive evaluation scope:** The benchmark includes a large number of subtasks spanning both textual and visual modalities, and the paper evaluates a diverse set of embedding models, enabling a broad comparison of model strengths and weaknesses across agent-relevant tasks.

I think the main weakness is that the paper does not evaluate some of the stronger multimodal embedding models. For example, if we refer to the MMEB leaderboard—a widely used benchmark for multimodal embeddings—most of the models evaluated in this paper do not rank among the top performers. It would be beneficial to include stronger and more recent models in the evaluation. Additionally, some of the models used appear to be outdated; for instance, VLM2Vec is evaluated using a relatively old checkpoint.

---

> ### Author Rebuttal · Authors · 2026-03-31
>
> Thank you for pointing out the need of recent multimodal models. We add evaluations of some recent models, Qwen3-VL-Embedding-2B and VLM2Vec-2.0, and will continue to expand our comparisons to include more newer models on MMEB.
>
> The results are presented in the table below. As shown, Qwen3-VL-Embedding indeed demonstrates stronger performance.
>
> | Model   | Context | Size | Rank_Borda | Avg.Type | Avg.66 | Deep QA19 | Tool4 | Code10 | Reasoning21 | Memory12 |
> |---------|--------|------|------------|----------|--------|-----------|-------|--------|-------------|----------|
> | VISTA   | 512    | 0.2B | 6          | 30.05    | 24.12  | 14.01     | 33.15 | 38.21  | 15.05       | 49.81    |
> | VLM2Vec | 8192   | 4B   | 7          | 20.45    | 18.16  | 24.33     | 17.64 | 14.29  | 6.33        | 39.65   |
> | GME-2B   | 8192   | 2B   | 4          | 42.64   | 39.22  | 48.65     | 40.21 | 56.88  | 18.33       | 49.12    |
> | GME-7B   | 8192   | 8B   | 2          |  /45.07/ |  /41.34/  | 50.42  | **42.15** | 62.34 |  /19.20/  | /51.24/  |
> | mmE5    | 8192   | 11B  | 5          | 35.33    | 33.23  | 44.08     | 34.39 | 50.55  | 15.28       | 32.34    |
> | Jina-v4 | 8192   | 3B   | 3          |  /45.07/     | 41.28  |  /51.08/  | 38.55 | **71.72** | 15.79 | 48.21 |
> | Qwen3VL-2B | 8192   | 2B   | 1          | **48.06**    | **44.54**  | **53.44** |  /40.45/  |  /70.48/ | **20.95** | **54.97** |
> | VLM2Vec2 | 8192   | 2B   | 8          | 16.05    | 14.19  | 13.12 | 9.27 | 29.36 | 5.50 | 22.99 |
>
> /xx.xx/ denotes the second-best results.
>
> Regarding the VLM2Vec results, we verified that our implementation produces aligned outputs with the official code under identical input examples. However, the reason for the lower evaluation scores remains unclear, and we are actively investigating this issue.

---

> > ### Author Rebuttal · Reviewer_GMjy · 2026-04-04
> >
> > The authors have provided results for several improved MLLM-based embedding models in the rebuttal; however, these updated results should be included in the analysis section of the paper.
> > For example, some of the results above look quite strange to me. Certain models perform much better than others on existing benchmarks, but the results here seem counterintuitive. It is understandable that models may perform differently across different task categories, but I think this deserves a clearer and more comprehensive analysis.
> > Overall, I would like to maintain my evaluation of this paper as borderline, with a slight lean toward rejection.

---

> > > ### Author Response · Authors · 2026-04-06
> > >
> > > Thank you very much for your feedback. We agree that some results are counterintuitive, e.g., VLM2Vec2.
> > >
> > > Upon further investigation after the initial rebuttal, we identified an issue in the MTEB implementation for VLM2Vec, where dataset-specific instructions were not properly used and a fixed generic instruction was applied instead. This led to a *significant underestimation of VLM2Vec2 performance*. We will contribute the revised code to MTEB to reflect the true potential of the VLM2Vec series models.
> > >
> > > https://github.com/embeddings-benchmark/mteb/blob/main/mteb/models/model_implementations/vlm2vec_models.py#L229
> > >
> > > After correcting this issue and re-running the evaluation, **VLM2Vec2 shows substantially improved results**, surpassing even larger models such as mmE5 (updated results are provided below). And VLM2Vec also presents better results.
> > >
> > > | Model   | Context | Size | Rank_Borda | Avg.Type | Avg.66 | Deep QA19 | Tool4 | Code10 | Reasoning21 | Memory12 |
> > > |---------|--------|------|------------|----------|--------|-----------|-------|--------|-------------|----------|
> > > | VISTA   | 512    | 0.2B | 7          | 30.05    | 24.12  | 14.01     | 33.15 | 38.21  | 15.05       | 49.81    |
> > > | VLM2Vec | 8192   | 4B   | 8          | 25.33    | 23.19  | 26.48     | 17.91 | 28.69  | 11.35        | 42.25   |
> > > | GME-2B   | 8192   | 2B   | 4          | 42.64   | 39.22  | 48.65     | 40.21 | 56.88  | 18.33       | 49.12    |
> > > | GME-7B   | 8192   | 8B   | 2          |  /45.07/ |  /41.34/  | 50.42  | **42.15** | 62.34 |  /19.20/  | / 51.24/  |
> > > | mmE5    | 8192   | 11B  | 5          | 35.33    | 33.23  | 44.08     | 34.39 | 50.55  | 15.28       | 32.34    |
> > > | Jina-v4 | 8192   | 3B   | 3          |  /45.07/     | 41.28  |  /51.08/  | 38.55 | **71.72** | 15.79 | 48.21 |
> > > | Qwen3VL-2B | 8192   | 2B   | 1          | **48.06**    | **44.54**  | **53.44** |  /40.45/  |  /70.48/ | **20.95** | **54.97** |
> > > | VLM2Vec2 | 8192   | 2B   | 6          | 36.59    | 34.10  | 42.65 | 26.77 | 53.17 | 14.01 | 46.36 |
> > > ||
> > > | VLM2Vec2 (old)| 8192  | 2B  | |16.05|14.19|13.12|9.27|29.36|5.50|22.99|
> > > | VLM2Vec (old)| 8192   | 4B   | | 20.45    | 18.16  | 24.33     | 17.64 | 14.29  | 6.33        | 39.65   |
> > >
> > > For the analysis, we observe that retrieval-specialized models (e.g., GME) tend to outperform general-purpose multimodal embedding models (e.g., mmE5, VLM2Vec) on AOEB, which focuses purely on retrieval tasks. This is consistent with the design trade-off that general embedding models aim to support a broader range of tasks (e.g., classification, grounding, retrieval), potentially sacrificing peak performance on a single retrieval task.
> > > For example, VLM2Vec2 exhibits anomalously low performance on textual tool retrieval (e.g., 31.12 vs. 45.00 for GME on ToolRet-Web), which aligns with its training data being oriented toward general-purpose embedding tasks. A similar trend is also observed in other general embedding models, such as mmE5 (34.24 on ToolRet-web).
> > >
> > > Now the new VLM2Vec-V2 is better than VLM2Vec, which aligns the trend on MMEB. We therefore believe that the updated results are now consistent with expectations and *no longer counterintuitive*.
> > >
> > > We will include the results and more analysis in revision.
> > > We suppose it could be addressed with minor revision, and do not change the overall conclusions of the paper.
> > >
> > > We hope these clarifications could address your concern. Thank you for your time!

---

### Official Review · Reviewer_5MPB · 2026-03-15

**Soundness:** 3
**Presentation:** 3
**Significance:** 3
**Originality:** 3
**Overall Recommendation:** 4
**Confidence:** 4

**Summary:**

The paper proposes AOEB, an embedding benchmark for evaluating embedding models’ retrieval capabilities for assisting agentic tasks. This paper is timely and assesses embedding models on realistic tasks that are more relevant today, and advocates moving beyond the common practice of overfitting to general embedding benchmarks.

**Compliance With Llm Reviewing Policy:**

Affirmed.

**Key Questions For Authors:**

1. Can the authors address questions above in the weakness part?
2. Do the authors have concrete suggestions (e.g., training, architectural) for the field based on the benchmark’s findings?

**Limitations:**

yes

**Strengths And Weaknesses:**

Strengths:

1. The categorization of this benchmark is nicely conceptualized. The included task categories cover essential capabilities for LLM agent workflows, including tool, deep QA, code, memory and reasoning retrieval.
2. The benchmark reveals critical capability gaps in current embedding models, and advocates deviating from the common practice of overfitting to general embedding benchmarks.
3. In general, the authors show a great understanding of the field, solid implementation of tasks and models. They provide comprehensive analyses including comparison with existing benchmarks and ablations of techniques used to cast interleaved and multi-image tasks as single-image tasks.

Weaknesses:

1. The distinction between Deep QA and reasoning-intensive retrieval might be a bit blurry conceptually. For instance, deep QA tasks also inherently require complex reasoning. Also, it appears that multi-hop datasets like MuSiQue are evaluated with a single-hop retrieval setting. Curious what do the authors think about this setting in terms of reflecting real-world usage, where the LLM agents typically break down complex questions into multiple search queries rather than relying on a single embedding pass?
2. The paper criticizes overfitting general benchmarks. However, AOEB is also a collection of static, public datasets with some of them already being heavily optimized for by the community. What do the authors think about this and if there’s any empirical proof that optimizing for these new agentic capability tasks conflicts with old general benchmarks?
3. A few detail discrepancies in the paper. Will be good to correct: (1) Qwen3-Embedding and NV-Embed’s AOEB scores in Figure 5 don’t match with Table 3 (significantly lower in Figure 5), which makes the findings conflict with what’s said between line 363-367.
(2) GritLM is mentioned as an evaluated model in 3.1 but never appears in the results throughout the paper.
(3) For Figure 5 upper part, the trends for AOEB and MIEB-lite actually are well-aligned (VLM2Vec < mmE5 < GME-7B), which conflicts a little bit with what’s said in line 359-line 362.
4. For the experiments adapting multiple image tasks as single image tasks, does this setting pose resolution disadvantages for static-resolution models? For non-dynamic-resolution models, the images will be highly blurred after concatenation and resized by standard processors, making it hard to tell whether the low scores are due to a lack of visual reasoning capabilities or the destructed visual details. Have the authors tried other techniques such as taking an average over single image embeddings etc?

---

> ### Author Rebuttal · Authors · 2026-03-31
>
> We greatly appreciate your insightful comments which could be helpful to improve our work, we address the concerns below.
>
> ### Q1
> > Distinction between Deep QA and reasoning. The multi-hop dataset
>
> We agree that reasoning is a general involved in many tasks, including Deep QA. We believe the distinction is not based on whether reasoning exists, but rather on the nature of the query–evidence relationship in retrieval. Specifically, Deep QA retrieval tasks focus on retrieving documents that contain *explicit or directly relevant evidence*, where the query and documents are relatively aligned semantically. While answering may involve reasoning, the retrieval step mainly requires identifying evidence-bearing documents. In contrast, reasoning-intensive retrieval tasks involve cases where relevant evidence is not explicitly aligned with the query, requiring implicit inference or abstraction to bridge the gap. This poses a different challenge for embedding models, which must capture latent relationships beyond surface-level matching.
>
> For the multi-hop, they are widely used in agentic RAG studies. In iterative retrieval process, some agents may perform pre-retrieval before action. The initial retrieval step is critical, as it determines the starting point of the reasoning process. And if the pre-retrieval results are strong, it could reduce the number of subsequent interaction rounds and thus improve overall efficiency.
>
> ### Q2
> > Overfitting and tasks conflicts
>
> Most datasets in AOEB contain only testset and no trainset, which reduces the likelihood of overfitting. Since much of the overfitting observed on general benchmarks is a result of directly training on the corresponding trainsets. It is true that certain tasks such as reasoning have recently seen substantial optimization, often through data synthesis or other strategies we think are beneficial.
>
> To mitigate overfitting risks:
>  - *from static to dynamic*: We believe that continuously updating the benchmark collection is an effective way to reduce overfitting. At task side, new tasks can be introduced over time, such as incorporating data related to MCP or skill retrieval. At data side, new samples can be constructed periodically based on emerging events and scenarios.
>  - *public and private*: Another potential approach is to reserve a portion of the benchmark as private test sets. These can be evaluated through a hosted server without releasing the data. New models would be evaluated on both the public and private splits; a significant performance gap on the private set may indicate potential overfitting.
>
> So far, we have not observed such evidence of confilct. In future work, we plan to explore to train embedding models that possess both strong general and agentic performance, as well as analyze how perf. interacts or confilct across tasks.
>
>
> ### Q3
> > A few detail discrepancies
>
> We apologize for the issues here. These discrepancies are mainly caused by incorrect Figure 5 version. We believe the main findings remain valid, and we will correct these issues in the revision.
> 1. The wrong version of Figure 5 was mistakenly included. While Qwen3-Embedding is indeed the strongest model, NV-Embed still underperforms compared with the other models, so the corresponding finding remains correct.
> 2. We forgot to remove it. It was originally planned, we will try it in revision.
> 3. We will revise the text. But we believe the finding is still reasonable because MIEB includes many tasks beyond retrieval, as well as multilingual components, which naturally leads to different trends than AOEB.
>
>
> ### Q4
> > Multi-image.
>
> This concat indeed introduces disadvantages for static-resolution models. This also reflects their inherent limitation in processing fine-grained information in image.
> Your example might be not well-suited for unified models, because text and images are tokenized and encoded through a shared backbone, so here is no seperated embeddings for each image.
> We try another *ensemble* method, averaging over {embed(text, img_1), embed(text, img_2), ...}. The results are shown in the table. It can be seen that it is comparable to merge-image (mi).
>
> ||traffic-mi|-ensemble|-vd|theorem-mi|-ensemble|-vd|
> |-|-|-|-|-|-|-|
> |GME-2B|33.12|33.20|37.54|25.43|25.45|26.45|
> |Jina-v4|15.33|15.30|25.14|19.84|19.38|21.38|
> |Qwen3VL-2B|44.57|44.85|44.30|33.95|33.94|30.10|
> |VISTA|21.53|21.12|6.69|19.93|19.94|0.98|
>
> ### Q5
> > Suggestions for the field
>
> - Task-specific optimization, such as training data synthesis, could be beneficial. The strong coding performance of jina-v4 suggests that targeted data optimization can significantly improve capabilities in certain domains.
>
> - Recent results from LightOnIO and MixedbreadAI indicate that ColBERT-style models perform very well on BrowseComp+ and other multimodal agentic tasks. We believe that late‑interaction architecture possess stronger compositional representation abilities, enabling them to handle complex retrieval tasks more effectively.

---

> > ### Author Rebuttal · Reviewer_5MPB · 2026-04-06
> >
> > Thanks for the clarification and the additional image ablation experiment. I believe that addressing the previously unclear points and data discrepancies mentioned above will strengthen the paper. The dynamic updates of the benchmark to prevent overfitting and the study of conflict between AOEB and general embedding benchmarks are planned as future work. I maintain my weak accept assessment.

---

> > > ### Author Response · Authors · 2026-04-07
> > >
> > > Thank you very much for your positive feedback. We are glad that the clarifications and additional experiments helped address the main concerns.
> > >
> > > Regarding the potential conflict between AOEB and general benchmarks: on the evaluation side, we will include additional evaluation and analysis of recently proposed agent-oriented text retrievers to better characterize this trade-off. On the modeling side, it is a future work to investigate this conflict by training state-of-the-art multimodal embedding models on AOEB.
> > >
> > > We believe these additions from your suggestions can be incorporated with minor revisions and will further strengthen the paper. Thank you again for your time and constructive comments.

---

### Decision · Program_Chairs · 2026-04-30

**Decision:**

Accept (regular)

**Comment:**

This well-presented work provides a benchmark for agent-based retrieval for embedding models, extending the existing relatively rich space of benchmarks for embeddings; algorithmic contributions are not a primary focus. Partly as a result, the novelty is limited; the benchmark consists of five newly reformulated datasets, one re-annotated dataset, and eleven existing but downsampled datasets. The overall design splits the benchmark into five task categories, some of which have some potential for overlap. The rebuttals were thorough and involved conducting new experiments, with additional results promised for the final version of the paper. One of the contributions claimed is an effort to continuously update the released benchmark in order to partially address the field's ongoing problem with benchmark overfitting; if this is performed, the contribution will be meaningful, although this is of course hard to evaluate with a single paper, which represents a static snapshot. On the positive side, the paper is well written, the rebuttals are thorough and address some of the concerns raised (I find the final categorization of concerns-raised and the claim that all are resolved somewhat oversimplified), and the benchmark has the potential to be of value to the community.